# CALOTRITON: A convective boundary layer height estimation algorithm from UHF wind profiler data

Alban Philibert[1,2], Marie Lothon[2], Julien Amestoy[3], Pierre-Yves Meslin[1], Solène Derrien[2], Yannick Bezombes[2], Bernard Campistron[2], Fabienne Lohou[2], Antoine Vial[2], Guylaine Canut-Rocafort[4], Joachim Reuder[5], and Jennifer Brooke[6]

[1]Institut de Recherche en Astrophysique et Planétologie, Université de Toulouse, CNRS, UPS, France
[2]Laboratoire d'Aérologie, Université de Toulouse, CNRS, UPS, France
[3]CEA, DAM, DIF, F-91297, Arpajon-Cédex, France
[4]CNRM-Université de Toulouse, Météo-France/CNRS, Toulouse, France
[5]Geophysical Institute, and Bergen Offshore Wind Centre, University of Bergen, and Bjerknes Center for Climate Research, Bergen, Norway
[6]Met Office, FitzRoy Way, Exeter, EX1 3PB, UK

**Correspondence:** Marie Lothon (marie.lothon@aero.obs-mip.fr)

**Abstract.**

Long time series of observations of atmospheric dynamics and composition are collected at the French Pyrenean Platform for the Observation of the Atmosphere (P2OA). Planetary boundary layer depth is a key variable of the climate system, but it remains difficult to estimate and analyse statistically. In order to obtain reliable estimates of the convective boundary layer height ($Zi$) and to allow long-term series analyses, a new restitution algorithm, named CALOTRITON, has been developed. It is based on the observations of a Ultra High Frequency (UHF) radar wind profiler (RWP) from P2OA, with the help of other instruments for evaluation. $Zi$ estimates are based on the principle that the top of the convective boundary layer is associated with both a marked inversion and a decrease of turbulence. Those two criteria are respectively manifested by larger RWP reflectivity and smaller vertical velocity Doppler spectral width. With this in mind, we introduce a new UHF-deduced dimensionless parameter which weights the air refractive index coefficient with the inverse of vertical velocity standard deviation to the power $x$. We then search for the most appropriate local maxima of this parameter for $Zi$ estimates, with defined criteria and constraints, such as temporal continuity. Given that $Zi$ should correspond to fair weather cloud base height, we use ceilometer data to optimize our choice of the power $x$, and find that $x = 3$ provides the best comparisons. The estimates of $Zi$ by CALOTRITON are evaluated using different $Zi$ estimates deduced from radiosounding, according to different definitions. The comparison shows excellent results with a regression coefficient of up to 0.96 and a root mean square error of 71 m, close to the vertical resolution of the UHF RWP of 75 m, when conditions are optimal. In more complex situations, that is when the atmospheric vertical structure is itself particularly ambiguous, secondary retrievals allow us to identify potential thermal internal boundary layers or residual layers, and help to qualify the $Zi$ estimations. Frequent estimate errors are nevertheless observed, for example when $Zi$ is below the UHF RWP first reliable gate, or when the boundary layer begins its transition to a stable nocturnal boundary layer.

# 1 Introduction

## 1.1 Instrumental techniques for convective boundary layer retrieval

The convective boundary layer (CBL) depth ($Z_i$) is a key variable in the climate system, for its role in modulating energy, water and trace species exchange at the interface between surface and free atmosphere. For this reason, it has significant applications in air quality, numerical weather predictions, climate models, and in more applied sectors such as renewable energy production. There are challenges in understanding the role of the convective boundary layer over heterogeneous surface, in complex terrain, coastal areas, polar regions, for surface/atmospheric exchange, transport and mesoscale circulation; all of which require a comprehensive estimate of the CBL depth. Although, it remains difficult to accurately and exhaustively quantify in the real world both in terms of the spatial and temporal variability, due to its complexity.

Instrumental techniques for $Z_i$ retrieval are numerous, and have lead to an abundance of literature. Kotthaus et al. (2023) propose a recent overview of the CBL top detection measurement techniques, with an exhaustive description of their capabilities and limitations. Here we summarise the most relevant techniques applicable to this study.

There are several ways to identify $Zi$, based on its characteristic atmospheric processes, which can be used to define different observational techniques. They can be classified in three main approaches: (i) based on the thermodynamical processes, (ii) based on the turbulent processes, and (iii) those based on tracers. Figure 1 schematizes those various definitions, through the vertical profiles of key variables.

The thermodynamical approach considers $Zi$ as the height, from the surface, at which the summital inversion occurs, characterized by strong gradients of temperature and moisture (Fig. 1a, 1b). Several instrumental methods estimate $Zi$ based on this approach, e. g. :

- the detection of gradients of either potential temperature, relative humidity ($RH$) or water vapor mixing ratio (e. g. Hennemuth and Lammert, 2006).

- the detection of the maximum of relative humidity (Couvreux et al., 2016).

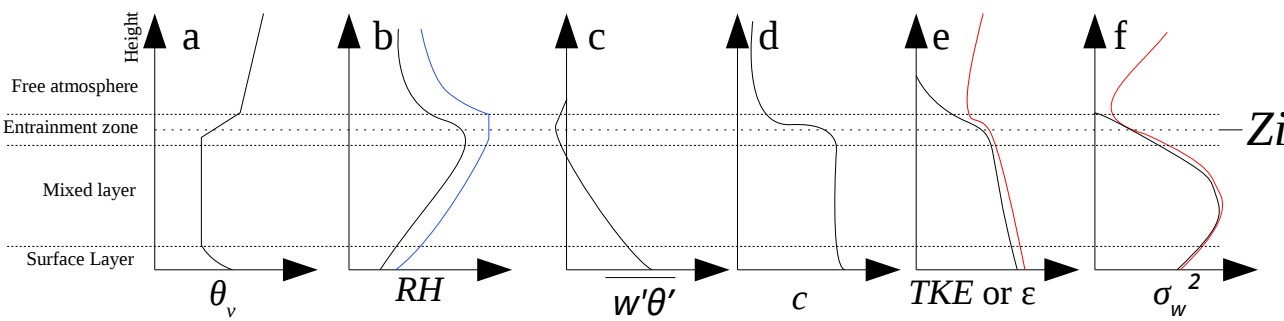

**Figure 1.** Idealized CBL profiles (black line) of (a) potential temperature, (b) relative humidity (blue line indicates the situation in the presence of clouds), (c) buoyancy flux, (d) scalar concentration, (e) turbulent kinetic energy ($TKE$) or turbulent dissipation rate ($\varepsilon$) and (f) vertical velocity variance (red line indicates the situation in the presence of external forcing).

- the so-called parcel method, which considers the potential temperature (or virtual potential temperature) at the surface $\theta_s$, and searches for the height above surface where $\theta = \theta_s$ (Holzworth, 1964), or $\theta = \theta_s + \delta\theta$, where $\delta\theta$ is a small positive variation of surface potential temperature (Seibert et al., 2000) .

In-situ measurements from radiosounding, aircraft or remotely piloted aircraft systems (RPAS) can be used, based on this approach. Remote sensing provides an indirect measure of thermodynamical variables, such as, Microwave Radiometers, Raman lidar, or differential absorption lidar. Indirectly related to this approach, Ultra High Frequency (UHF) radar wind profilers (RWPs), in L-band, are also appropriate devices to detect the CBL summital inversion, which is associated with a significant increase of reflectivity (White, 1993; Angevine et al., 1994).

Approaches based on turbulent processes consider $Zi$ as the height, from the surface, where turbulence intensity starts to strongly decrease (Fig. 1e). This height is coupled with minimum (and negative) buoyancy flux (Deardorff, 1972) and decrease of vertical velocity variance (Stull, 1988), Turbulent Kinetic Energy (TKE), or TKE dissipation rate ($\varepsilon$). Both buoyancy flux and vertical velocity variance reach zero above $Zi$, in textbook cases (unforced conditions, clear air). However in cases of external forcing such as clouds, wind shear or advection, a local minimum can be observed on each profiles (see red line Fig. 1e, 1f). Doppler lidar and UHF RWP give information on the turbulence intensity (Frehlich et al., 2006; Jacoby-Koaly et al., 2002, respectively). The variance of the Doppler velocity (Lothon et al., 2006), or the turbulent kinetic energy dissipation rate (e. g. Frehlich et al., 2006) can be used to detect the CBL top, based on a threshold. Note that studies based on numerical weather predictions models often use TKE as a reference for $Z_i$ determination (Couvreux et al., 2016), and studies based on Large Eddy Simulation often consider the minimum of buoyancy flux (e. g. Pino et al., 2006). This makes those turbulence-based approaches very relevant for model evaluation.

The tracer-based approach considers $Zi$ as the height, from the surface, where strong discontinuity is observed in the scalar concentration profiles (Fig. 1d) such as aerosol or gas concentration. Optical remote sensing, e.g. lidar and ceilometers, measure the optical backscatter coefficient from which aerosol concentration can be inferred (Kotthaus et al., 2023, for an exhausitive list). Wavelet methods are typically used to detect the top of the more loaded CBL (e. g. Haeffelin et al., 2012), where the aerosol concentration abruptly falls from the CBL to the free troposphere (see e. g. Davis et al., 2000, for the use of the Haar-wavelet-based method). The mixing ratio maximum gradient method described above could also be considered as a scalar concentration approach.

Some approaches use the synergy of instruments or methods. The bulk Richardson method (Hanna, 1969), with a threshold on the gradient Richardson, is a combination of wind gradient and potential temperature gradient. The complementarity of instruments is widely used for $Z_i$ estimations. For example, Min et al. (2020) or Turner and Lohnert (2021) use the association of a microwave radiometer with ceilometer or Raman lidar respectively. Since they are based on different definitions, all the methods discussed potentially result in slightly different estimates of $Zi$ (Caicedo et al., 2017), especially when the observed CBL is not a simple idealised case.

In this study, we revisit the methodology of estimating $Zi$ from UHF RWP measurements, and propose a new complementary algorithm. The advantage of RWP relatively to other remote sensing devices is their ability to measure in all weather types and not limited by cloud types and amount, precipitation or clear-sky conditions. Their height coverage limitation is predominantly

related due to water vapour content. One known weakness is their sensitivity to bird echoes, which typically occur at nighttime during bird migration events, particularly in spring and autumn. It is usually not a significant issue during daytime convection.

## 1.2 Motivations and main objectives

The multi-instrumented site of the ACTRIS-Fr[1] infrastructure, the Pyrenean Platform for the Observation of the Atmosphere (P2OA[2] – Lothon et al., 2023) gathers a comprehensive set of instruments for the monitoring of the atmosphere, with a sub selection of instrumentation located at the Center for Atmospheric Research (CRA), Campistrous, in South-West France, close to the Pyrenees mountain ridge. Among them, a UHF RWP purchased by the LAERO[3] laboratory has continuously measured the boundary layer since 2010. Retrieving the CBL height from this instrument from this multi-year time series allows a statistical study of the dynamical processes in this mountainous region. Processes include the influence of plain-mountain circulations, thermally-driven winds, the interaction between mountain waves and boundary layer, and the impact of mesoscale subsidence related to orographic convection. This unique dataset enables us to make statistical analysis and climatologies, with applications for air quality, weather forecasting and climate studies.

An existing technique, partly based on Angevine et al. (1994) was used for this specific radar for the estimate of $Zi$ (Jacoby-Koaly et al., 2002). Angevine et al. (1994) base the estimate of $Z_i$ on the absolute maximum of the air refractive index structure coefficient ($C_n^2$) which however does not always correspond to the current CBL top, but can correspond to a residual inversion above. To address this, Jacoby-Koaly et al. (2002) attempted to retrieve the local maximum of $C_n^2$ that could be the most appropriate estimate of $Z_i$, by use of temporal continuity, and other criteria. This is also the approach of Collaud Coen et al. (2014). Note that $C_n^2$ reaches local maxima where temperature and humidity show large vertical gradients, but also large wind gradients and turbulence (which induces fluctuations of air refractive index). This technique provides very satisfying results on a case-by-case investigation for fair weather convective conditions without complex vertical structure of the atmosphere (Heo et al., 2003; Jacoby-Koaly, 2000). However, statistical studies of the time series based on this technique may not be possible without significant errors. One obvious limitation for example, is that it often catches the top of the residual layer in early morning, rather than the top of the shallower developing CBL top. The temporal continuity criteria does not solve this issue. Attributing $Z_i$ as the top of the residual layer during the morning transition potentially leads to large errors. This can also occur in late afternoon, when this method will likely attribute $Z_i$ at the top of the pre-residual layer (Nilsson et al., 2016b) and then residual layer, while true $Z_i$ may decay with decreasing surface heat flux and decaying turbulence layer (Grimsdell et al., 2002; Lothon et al., 2014). It can also catch upper inversions, which are not directly connected to the mixed layer. Also note that residual layers are not always a local phenomenon, but may be advected (Angevine, 2000). In the presence of clouds at different levels, the difficulties increase due to more complexity, with greater stratification of the atmosphere and in cloud turbulence (Grimsdell and Angevine, 1998; Angevine, 2000; Collaud Coen et al., 2014; Duncan et al., 2022).

Several other techniques are based on the same principle of the existence of a local maximum of reflectivity. For example, Liu et al. (2019) uses local maxima of normalized Signal-to-Noise Ratio (SNR); Compton et al. (2013) uses the Covariance

---

[1]ACTRIS-Fr is the French component of the European Aerosol, Cloud and Trace Gases Research Infrastructure (ACTRIS), https://www.actris.fr/
[2]https://p2oa.aeris-data.fr/
[3]Laboratoire d'Aérologie, Université de Toulouse, CNRS, UPS, France

Wavelet Transform to detect the large step in SNR associated with $Z_i$; Molod et al. (2015) uses a simple algorithm with use of SNR, based on the determination of the "emergence time" and corresponding height, and temporal continuity based on backscatter standard deviation. All of them, however, encounter the same difficulties mentioned above, to more or less extent. Molod et al. (2015) used this technique to retrieve long series of $Z_i$ from a network of profilers, but the departure from in-situ estimates based on the bulk Richardson number shows that although simple and significantly robust, the proposed algorithm

still cannot handle the high complexity of the low troposphere. Collaud Coen et al. (2014) developed a climatology of the CBL height based on multiple remote sensing instruments, and validated the dataset against radiosoundings. They found that their estimates from the RWP were more dispersed, due to false attributions, revealing the difficulty of this approach to deal with the various encountered conditions.

     One way to improve this method is to also consider the decrease of turbulence at the top of the convective layer (see Fig. 1e,

1f), combined with the association of a local maximum of $C_n^2$ (Heo et al., 2003) or SNR (Bianco and Wilczak, 2002; Bianco et al., 2008). Heo et al. (2003) assume that the zero buoyancy flux is reached where the vertical velocity standard deviation is null. They search for this height, and then select the nearest local maximum of $C_n^2$. With the same basic assumption, Bianco and Wilczak (2002) and Bianco et al. (2008) use fuzzy-logic method to determine the height corresponding to the combination of radar variables. Those techniques definitely improve the CBL depth estimates, relatively to the more standard approaches.

In this study, we use this same fundamental assumption and combination to improve the initial method used for the LAERO UHF RWP radar, in order to develop a new algorithm to address a broader range of atmospheric conditions, including complex vertical structure of the atmosphere, cloudy situations and multi-layered lower troposphere.

     We present the experimental data used in Sect. 2, describe the $Zi$-retrieval algorithm (CALOTRITON) and discuss the choice of configuration parameters in Sect. 3. Illustrative examples are given in section. 4, and a comparison of the CALOTRITON

UHF-based $Zi$-estimates to in-situ measurements is presented in Sect. 5. A conclusive discussion is drawn in Sect. 6.

## 2    Instrumentation and data

### 2.1    Datasets

In this study, we consider the data of the LAERO UHF RWP at the P2OA-CRA from 2015 to 2022, to develop the new CALOTRITON algorithm. This time period is shorter than the whole available time series, for sake of data process homo-

geneity. The year 2018 is more intensively used for the primary development of the algorithm. We also use sensible heat flux measurements from a sonic anemometer installed at 30 m on the P2OA-CRA 60 m instrumented tower, and relative humidity measurements, made at 2 m.

     To optimize CALOTRITON parameters, we compare $Zi$ estimates with cloud base heights (Sect. 3.3) measured by a CT25k ceilometer from Centre National de Recherche Météorologique (CNRM), installed from December 2016 to December 2019 at

P2OA-CRA.

The algorithm results are then validated (see Sect. 5) by comparison to in-situ profiles made with radiosonde or Remotely Piloted Airplane System (RPAS) during two intensive field campaigns; (i) BLLAST (Boundary Layer Late Afternoon and Sunset Turbulence, Lothon et al., 2014), which took place at P2OA-CRA and (ii) LIAISE (Land surface Interactions with the Atmosphere over the Iberian Semi-arid Environment, Boone et al., 2021), which took place in North-East Spain, close to Lleida. The latter enables us to test CALOTRITON in another meteorological and geographical context to that of the long term observational record of P2OA-CRA, and thus generalize its applicability and use. For both measurement campaigns, the CNRM UHF RWP is used in addition to the LAERO UHF RWP which also enables to test the algorithm on a different UHF RWP.

Table 1 summarises the contexts of RWP measurements used and corresponding time period, the location of RWPs, the complementary instrumentation used and their role in this study. The corresponding datasets are listed and referenced in the *Data availability* section, at the end of the article, with more precision on the specific periods for each instrument.

**Table 1.** Summary of instruments used and context

| Context | Period | RWP | RWP Location | Complementary instrumentation | Use for CALOTRITON |
|---|---|---|---|---|---|
| P2OA | 2015-2022 | LAERO UHF RWP | Campistrous, France | sonic anemometer CSAT3 | optional input |
| | | | | Humidity sensor HMP45 | input |
| | | | | CT25k Ceilometer | configuration optimization |
| | | | | Radiosoundings | validation |
| BLLAST | June - July 2011 | LAERO UHF RWP | Campistrous, France | sonic anemometer CSAT3 | optional input |
| | | | | RPAS | validation |
| | | | | Radiosoundings | validation |
| | | CNRM UHF RWP | Capvern, France | RPAS | validation |
| | | | | Radiosoundings | validation |
| LIAISE | July 2021 | LAERO UHF RWP | Els Plans, Spain | sonic anemometer | optional input |
| | | | | Radiosoundings | validation |
| | | CNRM UHF RWP | La Cendrosa, Spain | sonic anemometer | optional input |
| | | | | Radiosoundings | validation |

The sensible heat flux is calculated on 30 min duration samples with EddyPro software, based on Eddy-Correlation technique. The UHF RWP instruments and data process is detailed in the following.

## 2.2 UHF radar wind profiler technical characteristics and data process

The LAERO UHF RWP is 1.274 GHz wind profiler with 5 beams, four oblique beams and one vertical beam. Its main characteristics are listed in Table 2 (for more details, see Jacoby-Koaly, 2000). It runs alternatively with two modes: one 'low mode' with a pulse width of 500 $ns$ corresponding to a range resolution of 75 m, and a 'high mode' with the pulse width of 2.5 $\mu s$ corresponding to a range resolution of 150 m and a slightly better height coverage. For our use here, we only consider the low mode. The maximum height for this mode is usually around 3 km a. g. l., but may be only 500 m or 1000 m in winter, when dry anticyclonic conditions occur. It can reach 7 or 9 km within deeper clouds and rain. The first gate is 75 m, but with a poor confidence index. We consider the 225 m gate as first gate with very good confidence. The CNRM UHF RWP mainly presents the same characteristics but with a first level with a good confidence index at 300 m.

**Table 2.** Main characteristics of the LAERO UHF RWP (https://p2oa.aeris-data.fr/sedoo_instruments/profileur-de-vent-uhf/).

| | |
|---|---|
| Manufacturer | Degreane |
| Reference | PCL1300 |
| Emission frequency | 1.274 GHz |
| Number of beams | 5 (N-W-S-E-vertical) |
| Transmission Frequency | 1274 MHz |
| Opening Angle | 8.5° |
| Obliques antennas Angle | 17° to the vertical |
| Vertical Resolution | 75 m |
| Temporal Resolution | $\sim$ 2 min |
| First level with a good confidence index | 225 m |
| Vertical coverage | $\sim$ 3 km |

The 3 components of the wind are deduced from the Doppler radial velocity of the 5 beams, every 75 m, and every 2 minutes. The first main critical step is to select the meteorological peak from the Doppler spectrum. We use a process developed at LAERO laboratory, which optimize the meteorological peak selection and data coverage, relative to the manufacturer processing. During this phase, an automatic quality control is done, eliminating Doppler spectral erroneous peaks before the wind component calculation. The second step is typical of velocity volume processing technique (Wadteufel and Corbin, 1979), which computes the three wind components from the radial velocity, with minimum least square error. The air refractive index structure coefficient $C_n^2$ is deduced from the reflectivity as a function of the received power (Doviak and Zrnic, 1993). The vertical velocity variance $\sigma_w^2$ is obtained from the spectral half Doppler width of the backscattered signal on the vertical beam, and gives and allows to estimate TKE dissipation rate $\varepsilon$ (Cohn and Angevine, 2000; Jacoby-Koaly et al., 2002). Hereafter, $C_n^2$ is the median air refractive index structure coefficient over the 5 beams, and depends on altitude and time. $\varepsilon$ is the median TKE dissipation rate over the 5 beams. $\sigma_w$ is deduced from vertical antenna and corrected for the effect of the horizontal wind within the antenna aperture (Jacoby-Koaly et al., 2002). All those variables are calculated at 2 min time interval.

## 3 The CALOTRITON algorithm

### 3.1 CALOTRITON specific objectives

The new $Zi$-retrieval algorithm (CALOTRITON) was developed with 5 main objectives and constraints:

1. To restrict $Zi$ estimate to the convective boundary layer, by only considering daytime conditions and excluding precipitation periods.

2. To respect temporal continuity of $Zi$ growth and to follow it as finely as possible in time, in order to describe the smallest convective scales (5 to 30 minutes, Stull, 1988). $Zi$ should start close to the ground early in the day.

3. To manage complex cases: as in the presence of clouds, or thermal internal boundary layer (TIBL), when cold air advection in the lower layers can create a new convective boundary layer e. g., in case of slope wind (Kossmann et al., 1998) or sea breeze (Durand et al., 1989).

4. To take into account abrupt CBL growth, which occurs in the presence of a residual neutral layer above Zi, when the current CBL potential temperature gets to reach the residual neutral layer potential temperature (Blay-Carreras et al., 2014).

5. To use limited instrumental synergy in order to apply it in other sites (or measurement campaigns) equipped with a UHF RWP, and not to depend on the availability of an advanced instrument suite to establish $Zi$ estimate.

## 3.2  CALOTRITON operation

Figure 2 presents a scheme of CALOTRITON algorithm which is described in this section, and Table 3 recapitulates the variables used at the different steps of CALOTRITON, with the corresponding time scale.

**Table 3.** Variables used in CALOTRITON, at the different steps of the algorithm, and their time interval

| Input variables | $C_n^2, \sigma_w, \varepsilon$ and $w$ | 2 min | Main input variables |
|---|---|---|---|
| | $H$ | 30 min | $t_{init}$ assessment (optional) |
| | $RH$ | 1 s | Fog occurrence estimation (optional) |
| **Filtered variables** | $C_n^2, \sigma_w, \varepsilon$ | 5 min | |
| **Calculated variables** | $NPx, Zi_\varepsilon$ | 5 min | Key intermediate variables |
| | $t_{init}$ | 1 day | Key CBL growth starting variables |
| **Auxiliary variable** | $CBH$ | 1 min | Configuration optimization |
| **Final variables** | $Zi_{NP3_{std}}$ | 5 min | Best estimate |
| | $Zi_{NP0_{std}}, Zi_{NP0_{sup}}, Zi_{NP0_{sub}}$ | 5 min | Complementary estimates |
| | $QF$ | 5 min | Quality assessment |

### 3.2.1  Restriction to CBL conditions

First, we consider UHF RWP data only above 225 m a. g. l., and below 3000 m a. g. l.. 225 m is the first gate where data is always of high quality. Only daytime data are selected to estimate $Zi$ from the UHF RWP. For this, sunrise and sunset times are retrieved as a function of date, altitude, latitude and longitude. Precipitation periods (including virga) are excluded by a function based on empirical thresholds on $C_n^2$ and Doppler vertical velocity ($w$). Any profile which meets $C_n^2 > 10^{-14}$ m$^{-2/3}$ and $w < $ - 1 m s$^{-1}$ over five consecutive levels is removed, as well as all profiles occurring 15 minutes before and after. We do not assign $Zi$ in case of fogs, notably due to the limitation of the UHF RWP below 225 m a. g. l.. It was found at P2OA site that relative humidity at 2 m larger than 90 % was associated to fog occurrence as confirmed by ceilometer measurements (not shown). We therefore take this as criterion for fog occurrence, and remove corresponding periods from the further analysis.

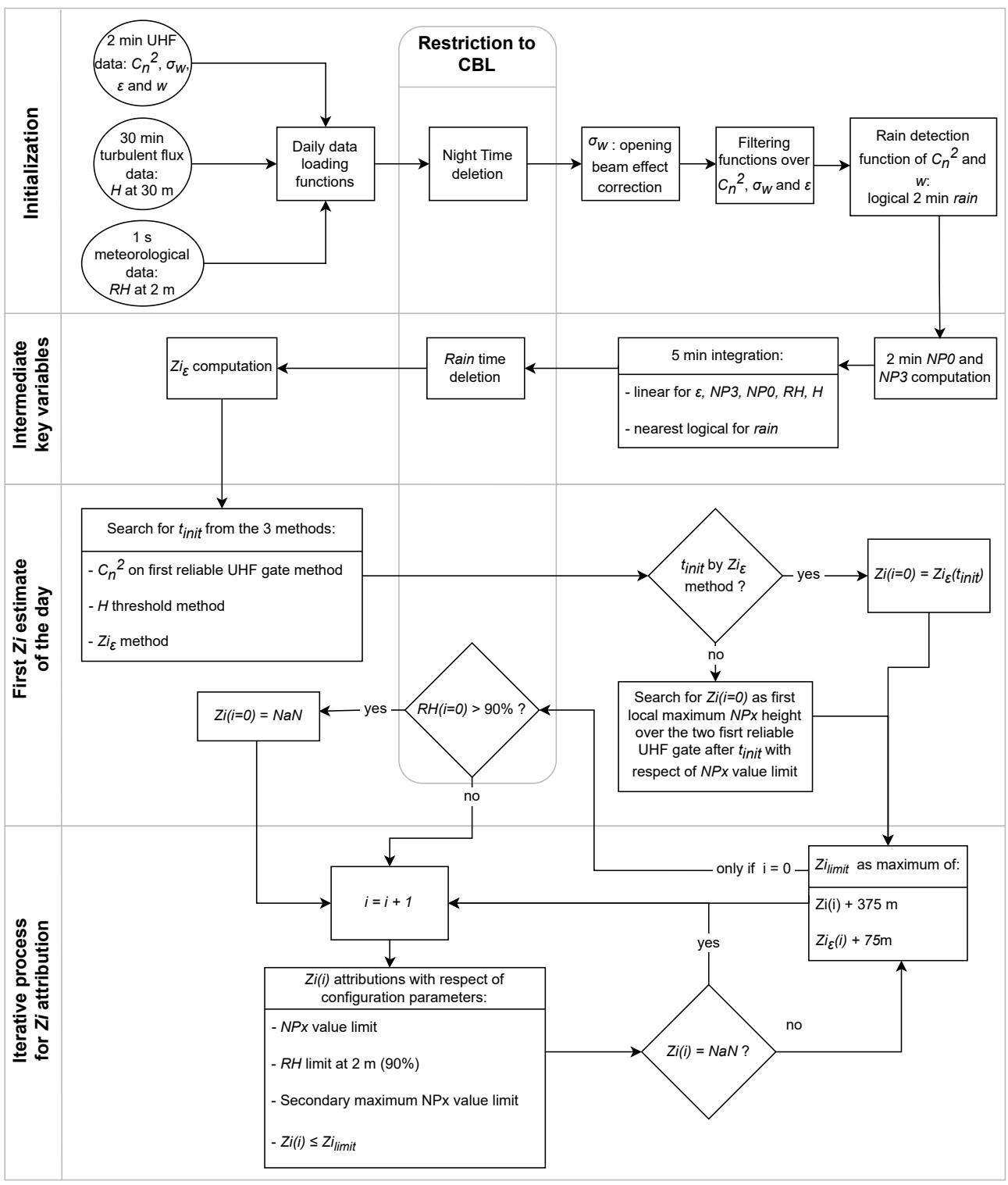

**Figure 2.** CALOTRITON organisation chart.

### 3.2.2 Data averaging

In order to disregard non-meteorological disturbances (e. g., birds) on the UHF RWP signal, $C_n^2$, $\sigma_w$ and $\varepsilon$ data are filtered by complementary sliding median filters:

- $C_n^2$ and $\varepsilon$: median over 6 minutes (3 points), none over height in order to keep the original UHF RWP vertical resolution of 75 m ($dz$).

- $\sigma_w$ : median over 8 minutes (4 points), and a median over 225 m (3 points), because of a more pronounced spatio-temporal variability of these data (see Fig. 4d, 6d 8d ). We use coarser filters for $\sigma_w$ to compensate for the fact that $C_n^2$ and $\varepsilon$ are already integrated over the 5 beams.

If a larger integrated time is chosen, the corresponding median time filter should be adjusted and applied to $C_n^2$, $\varepsilon$ and $\sigma_w$.

### 3.2.3 Definition of intermediate key variables

As the reflectivity maximum does not always correspond to $Zi$, especially in the case of a cloudy sky, we suggest using a new dimensionless variable which takes into account both, the increase of reflectivity at the summital inversion and the decrease of turbulence: $NPx$ (eq. 1) weights $C_n^2$ by $\sigma_w$ power $x$, and allows for a better account of a large value of $C_n^2$ associated with a small value of $\sigma_w$. Dimensionless $NPx$ is obtained by averaging values of $C_n^2$ and $\sigma_w^x$ up to 3000 m for each profile (overlines in eq. 1):

$$NPx = \frac{C_n^2/\overline{(C_n^2)}}{\sigma_w^x/\overline{\sigma_w^x}} \tag{1}$$

$NPx$ is computed with the filtered data discussed previously (section 3.2.2) and is linearly integrated over a 5 minute time step to describe the smallest characteristic convective scale. The choice of $x$ is discussed in the Sect. 3.3.2. As examples, Figures 4f and 6f discussed later, show cross sections of $NP3$ in a simple and complex case respectively. Note that this approach is based on the same main assumption as in the methods proposed by Heo et al. (2003) and Bianco and Wilczak (2002), who also combined the need of an increased reflectivity and a decrease of turbulence.

We also consider another variable, purely defined by the level of turbulence: $Zi_\varepsilon$ is the height above the surface at which the TKE dissipation rate $\varepsilon$ falls below $5 \times 10^{-4}$ m$^2$ s$^{-3}$. This technique was previously used by (Couvreux et al., 2016; Nilsson et al., 2016a). It thus represents a rough estimate of the depth of significant turbulence. $Zi_\varepsilon$ is computed on filtered and integrated $\varepsilon$ data (5 minutes as $NPx$). In order to consider only the $Zi_\varepsilon$ that would respect a certain temporal continuity, a sliding median filter over 15 minutes (3 points) is applied on $Zi_\varepsilon$.

$NPx$ is the core variable of CALOTRITON, but $Zi_\varepsilon$ will help us on documenting the associated turbulence, and optimize the selection of the most appropriate local maximum of $NPx$ as an estimate of $Zi$ (we hereafter call this selection "$Zi$ attribution").

### 3.2.4 Determination of the first $Z_i$ estimate of the day

In a typical CBL development, $Z_i$ starts close to the ground, below the UHF RWP detection limits (225 m), and grows until it reaches a plateau in the early afternoon (Stull, 1988). It is therefore necessary to wait for some time (called $t_{init}$) before $Z_i$ can be detected by the UHF RWP. We found that the sensible heat flux, which governs the evolution of $Z_i$, remains very low (less than a few tens of W) at least until an hour and a half after sunrise (not shown). Therefore, $t_{init}$ is not defined before 1.5 hour after sunrise.

Several methods are used to determine $t_{init}$. The first is based on $C_n^2$ at the first reliable UHF RWP gate (225 m a. g. l.) and considers $t_{init}$ as the time when the 30-minute sliding median exceeds its daily mean value. That way, it is investigated when an increase in $C_n^2$ becomes significant and may correspond to $Z_i$. The second method is based on the measured sensible heat flux ($H$) and considers $t_{init}$ when $H$ exceeds a significant threshold of 50 $Wm^{-2}$. $t_{init}$ is taken as the earliest time over those two. The first assigned $Z_i$ of the day ($Z_i(i = 0)$) can only be established at a local maximum of the vertical profile of $NPx$ located at one of the two first reliable levels of the UHF RWP and occurring after $t_{init}$.

Sometimes, a thin layer is mixed by dynamical turbulence before sunrise, e.g., in the presence of a low level jet. In order to take those situations into account, we allow the attribution of the first $Z_i$ at the height of $Z_{i_\varepsilon}$ if the latter corresponds exactly to the height of the $NPx$ maximum of the profile, independently of $t_{init}$, and provided that this attribution is always done 1.5 hour after sunrise.

This initialization process is somehow similar to Molod et al. (2015), who called this time the 'emergence time', and determined it based on the same principle, i. e. they also consider a first good confidence gate and look for the first determinable $Z_i$ at this level, but in a different way.

### 3.2.5 Iterative process for $Z_i$ attribution

Once the initial $Z_i$ is found, the search for subsequent $Z_i$ is done by temporal iteration on the most significant local maximum of $NPx$ that is located within a vertical growth limit of 375 m since its last effective attribution. Residual layers or clouds above $Z_i$ can potentially return a higher signal contribution to $NPx$ than $Z_i$ itself, and might be misinterpreted if located within the 375 m growth limit. To take this into account, the algorithm allows attributions on local secondary maxima of $NPx$ below the first if the value of the corresponding $NPx$ is at least 90 % of the first maximum value of $NPx$ before 10:00 UTC and 50 % after. These empirical values are discussed in section 3.3.2 and named relative thresholds of secondary maximum of $NPx$. Finally, a minimal value of $NPx$ is required for attribution and fixed to the mean profile value of $NPx$ in order to take into account a certain significance. Sometimes, strong growth of $Z_i$ can occur and exceed the imposed limit (375 m). This motivated us to use $Z_{i_\varepsilon}$, in order to consider up to which level significant turbulence is found. If at $i$ time, $Z_{i_\varepsilon}(i)$ is higher than the last effective attribution plus the growth limit, then $Z_i(i)$ can be searched up to $Z_{i_\varepsilon}(i) + dz$ (where $dz$ = 75 m).

### 3.3 Algorithm parameter choice

#### 3.3.1 Parameter optimization

All the parameters presented above were obtained empirically by subjectively judging the quality of the attributions of $Zi$ for about 100 days in 2018 at P2OA-CRA. In order to verify their quality in a more objective way and possibly to adjust some parameters, we compared the estimates of $Zi$ with the lowest cloud base height (CBH) measured by the CT25k ceilometer within a 5-minute interval around each attribution. This comparison is based on data from December 2016 to December 2019. When comparing two configurations with the distributions shown in Fig. 3, one would favour the configuration which leads to less attributions above cloud base and lower values of $\varepsilon$ at $Zi$.

Figure 3 shows an example of the results of this comparison, for $Zi$ estimates based on either $NP3$ or $NP0$, with the use of the optimal parameters listed in Table 4. Figure 3a shows the distribution of the set of $Zi$ attributions for the different $NPx$ ($x$=0 and $x$=3), and indicates more attributions by $NP3$, especially for $Zi < 700$ m. Figure 3b shows the distribution of the differences between $Zi$ and CBH. It can be seen that there are slightly more attributions above the cloud base when using $NP0$. Figure 3c presents the distribution of all $\varepsilon$ values at $Zi$ height, and shows that $NP3$ attributions tend to get lower $\varepsilon$ values at $Zi$ height. The fact that $NP3$ attributions of $Zi$ are more often lower than $NP0$ attributions and associated with lower $\varepsilon$ values is

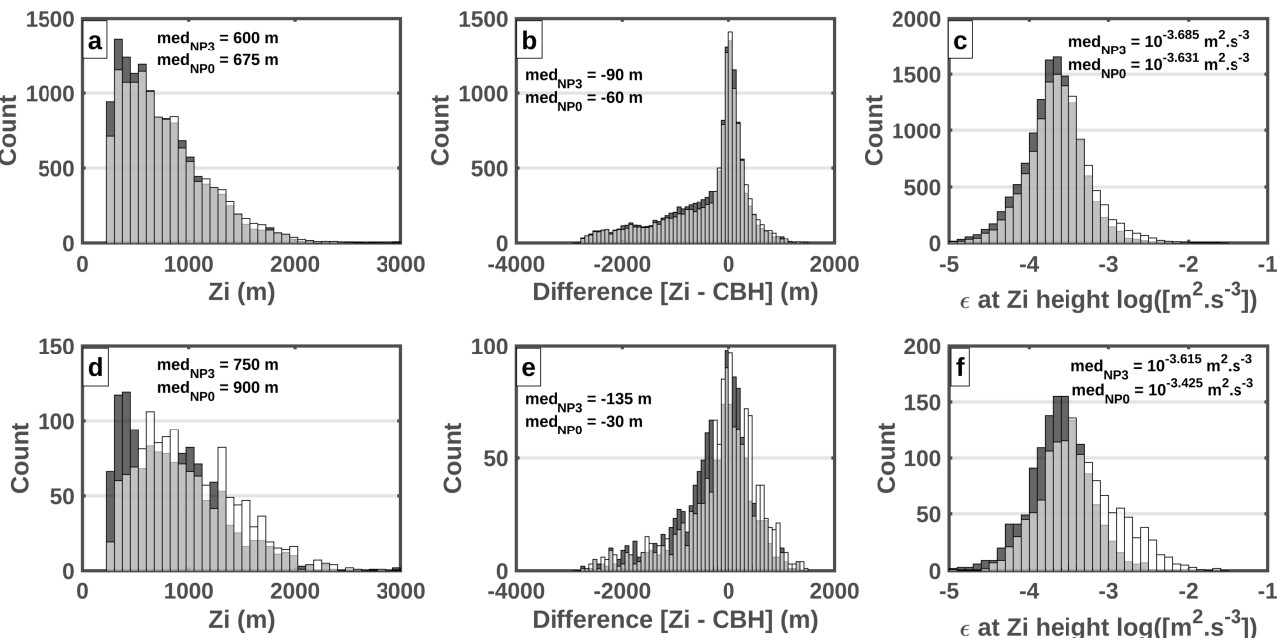

**Figure 3.** Histograms showing the differences between the distributions of $Zi_{NP0_{std}}$ (white bar) and $Zi_{NP3_{std}}$ (black bar) in presence of cloud measured by the CT25k ceilometer from December 2016 to December 2019: (a) $Zi$ distribution, (b) distribution of difference between $Zi$ and cloud base height; (c) $\varepsilon$ value distribution at $Zi$ height. (d) to (f) are respectively the same as (a) to (c) but considering only attributions which present more than 225 m difference between $Zi_{NP0_{std}}$ and $Zi_{NP3_{std}}$. For each distribution, the median values are indicated by $med_{NP3}$ and $med_{NP0}$, for distribution from $Zi_{NP3_{std}}$ and $Zi_{NP0_{std}}$ respectively.

a sign of better quality attributions. When clouds are present, the difference between $Zi$ estimates with $NP3$ and $NP0$ is on average twice as large as in clear sky cases, due to the complexity of the atmosphere in cloudy conditions. Thus, the observed differences between $Zi$ attributions with $NP3$ and $NP0$ give an indication of the CBL complexity. Figure 3d to 3f presents the same figure as the top panel (Fig. 3a to 3c) but only considering the attributions by $NP3$ and $NP0$ when they differ for more than 225 m from each other. This represents only 10 % of the total attributions. The same conclusions as previously stated can

be drawn, even more clearly here. We therefore confirm that $NP3$ statistically gives better results.

### 3.3.2   Tested parameters and optimum set

In this way, the set of $NPx$ for $x$ = 1 to 5 were compared two by two with the configuration presented in Table 4. It was noted that attributions were potentially better for $x$ = 3 rather than $x$ = 0, 1 or 2. However, no significant trend was noticed for $x \geq 3$. We limit us to $x$ = 3 in order to keep attributions predominantly based on $C_n^2$. In this section, only a few results of our search for

the best parameters by attribution distribution analysis are presented. All are based on $NP3$. The largest differences appeared between whether or not we consider a limit on relative humidity. Not setting a limit allows about 4% more attributions in clear sky and 40% more in the presence of clouds. Among those 40%, half of them corresponds to cloud base heights below 225 m, which is the first level of the UHF RWP. Considering the limit on relative humidity, 13% of all attributions in the presence of clouds take place 225 m above the cloud base, compared to 22% without a limit. This limit therefore both avoids attributions in

the presence of clouds whose base is below the UHF RWP lower limit and reduces the number of attributions above the cloud base by half.

    The methods for the search of $t_{init}$ were tested. Using solely the $C_n^2$ maximum technique leads to almost no difference in $Zi$ attributions, but additionally using the technique based on sensible heat flux leads to 3% more attributions.

    Other values related to the growth limit were also tested. It was noticed that a limit of 300 m with the last effective attribution

potentially allows to obtain better quality attributions but leads to a reduction of 3% of the attributions compared to a limit of 375 m. Empirically, it was found that 300 m was not sufficient to properly track the evolution of $Zi$ compared to 375 m.

**Table 4.** List of best parameters for CALOTRITON configuration

| Criterion number | Parameter | Value | Comments |
|---|---|---|---|
| ♯ 1 | Integration time | 5 minutes | |
| ♯ 2 | Time median filter $C_n^2$ | 3 points | ∼ 6 minutes |
| ♯ 3 | Time median filter $\varepsilon$ | 3 points | ∼ 6 minutes |
| ♯ 4 | Time median filter $\sigma_w$ | 4 points | ∼ 8 minutes |
| ♯ 5 | Height median filter $C_n^2$ | 0 points | 0 m |
| ♯ 6 | Height median filter $\varepsilon$ | 0 points | 0 m |
| ♯ 7 | Height median filter $\sigma_w$ | 3 points | 225 m |
| ♯ 8 | Growth limit | 375 m | between two effective assignments |
| ♯ 9 | Relative humidity limit at 2m | 90 % | |
| ♯ 10 | $NPx$ Value limits | $NPx$ profile mean | |
| ♯ 11 | Secondary maximum $NPx$ value limit | 90% before and 50% after 10:00 UTC | criterion ♯ 10 applied |
| ♯ 12 | $Zi_\varepsilon$ option | True | To exceed the growth limit |

On the other hand, a 450 m growth threshold did not improve statistically the results. Although it leads to an increase of the total number of attributions by 3%, all additional attributions under cloudy skies were above cloud base. This is the reason we finally chose 375 m as the optimal growth threshold.

Other important parameters are the values selected for the relative thresholds of secondary maximum $NPx$ on which attributions are possible. Not setting a limit leads to an increase of 40% in attributions above CBH + 225 m, associated with higher $\varepsilon$ values, which is thus less appropriate. Thresholds of 50% and 90% were tested over the whole day and it was found that 50% led to more attributions over residual layers than 90%, especially in the morning. In contrary, a threshold of 90% leads to more attributions inside the CBL, especially in the afternoon. This is why a threshold of 90% before 10:00 UTC and 50% afterwards
was chosen. A threshold of 75% for the whole day was also tested but provided poorer results.

### 3.3.3   Final assignment and flags

As we have seen previously, the difference between $NP0$ and $NP3$ attributions with the parameter set as described in Table 4, gives a useful and complementary information about the complexity of the lower troposphere. This is why we perform four estimates of $Zi$:

– $Zi_{NP3_{std}}$: estimated with standard configuration for $NP3$ as described in Table 4, considered as the best attributions.

– $Zi_{NP0_{std}}$: estimated with standard configuration for $NP0$ as described in Table 4.

– $Zi_{NP0_{sup}}$: estimated from $NP0$ as described in Table 4, but without applying criteria ♯ 9, 10, 11. With this configuration, the 375 m growth limit (♯ 8) is applied between the searched $Zi$ and the maximum $Zi$ already allocated. There is also no $t_{init}$ restriction after sunrise. This configuration allows to search for levels higher than the estimates made with a
standard configuration, which may correspond to the top of a residual layer, or to $Zi$ if the standard configurations assign on a TIBL top.

– $Zi_{NP3_{sub}}$: estimated from $NP3$ as described in Table 4, but without applying criterion ♯ 11. With this configuration, the $NPx$ profile mean of the criterion ♯ 10 is replaced by the median which gives lower values most of the time, mainly because of high values of $C_n^2$. This configuration allows us to search for levels lower than the estimates made with a
standard configuration, which may correspond to a TIBL top, or to $Zi$ if standard configurations assign on a residual layer top.

Our best proposed estimate is $Zi_{NP3_{std}}$, for the reasons explained before. But the four estimates embed the large complexity that is often observed in the lower troposphere.

In order to qualify this complexity and to facilitate the correct use of the four estimates, a quality flag $QF$ is defined :

– $QF$ = 1: all attributions are equal. It indicates a very good confidence in the assignment quality and a textbook case.

– $QF$ = 2: only $Zi_{NP3_{std}}$, $Zi_{NP0_{std}}$ and $Zi_{NP3_{sub}}$ are equal. It indicates a good confidence in the assignment quality and the likely presence of a residual layer above $Zi$, located at $Zi_{NP0_{sup}}$. It also indicates that the $Zi$ estimate does not match with the height of the $C_n^2$ maximum.

- $QF = 3$: only $Zi_{NP3_{std}}$, $Zi_{NP0_{std}}$ and $Zi_{NP0_{sup}}$ are equal. It indicates a medium confidence in the assignment quality and the likely presence of a TIBL located at $Zi_{NP3_{sub}}$.

- $QF = 4$: only $Zi_{NP3_{std}}$, $Zi_{NP0_{std}}$ are in exact agreement. It indicates a medium confidence in the assignment quality and the likely presence of both a TIBL and a residual layer, located at $Zi_{NP3_{sub}}$ and $Zi_{NP0_{sup}}$, respectively.

- $QF = 5$: no agreement between the four attributions of heights. This indicates poor confidence in the assignment quality, and a highly complex case.

Other flags could be produced, in order to more thoroughly document the meaning of those various estimates. They could for example qualify the temporal continuity of $Zi_{NP3_{std}}$ (occurrence of abrupt changes,...) or the consistency of $Zi_{NP3_{std}}$ with $Zi_{\varepsilon}$.

## 4  Illustrating case studies

In this section, we present three study cases to illustrate the capability of CALOTRITON, and the improvements of $Zi$ retrieval relatively to a more standard approach:

- A reference simple clear sky case (27 October 2021, at P2OA)

- A complex cloudy sky case (15 March 2018, at P2OA)

- A complex multiple layering clear sky case (27 July 2021, during the LIAISE field experiment)

### 4.1  Clear sky case at P2OA

Figure 4 shows the height-time section of four UHF-based variables defined before: the air refractive index structure coefficient $C_n^2$ (Fig. 4a), the air vertical velocity variance $\sigma_w^2$ (Fig. 4b), the turbulent kinetic energy (TKE) dissipation rate $\varepsilon$ (Fig. 4c), and the new combined parameter $NP3$ (Fig. 4d).

The downward short-wave radiation (white line) and the sensible heat flux (blue line) are overlaid on all panels. The short-wave radiation shows that this day was mainly clear, with only a few thin and occasional cirrus clouds in the afternoon. Sensible heat flux shows a typical diurnal cycle. Also overlaid are different estimates of $Z_i$, defined in the previous section: $Zi_{NP0_{std}}$, $Zi_{NP0_{sup}}$, $Zi_{NP3_{std}}$, $Zi_{NP3_{sub}}$ and the intermediate variable $Zi_{\varepsilon}$. We note for this case a very large consistency between the four different estimates $Zi_{NPx}$. That means whatever the method, standard or more sophisticated, taking account on turbulence intensity or not, they all agree for $Z_i$ estimation for the CBL growth, and simply match to the absolute maximum reflectivity for most of the time.

In order to make the correspondence between the UHF RWP and the thermodynamical profiles, Fig. 5 compares in-situ measurements of thermodynamical variables measured by radiosondes with the UHF RWP measured variables at 13:35 UTC, on this same clear day of 27 October 2021. The comparison shows that the absolute maximum reflectivity corresponds well

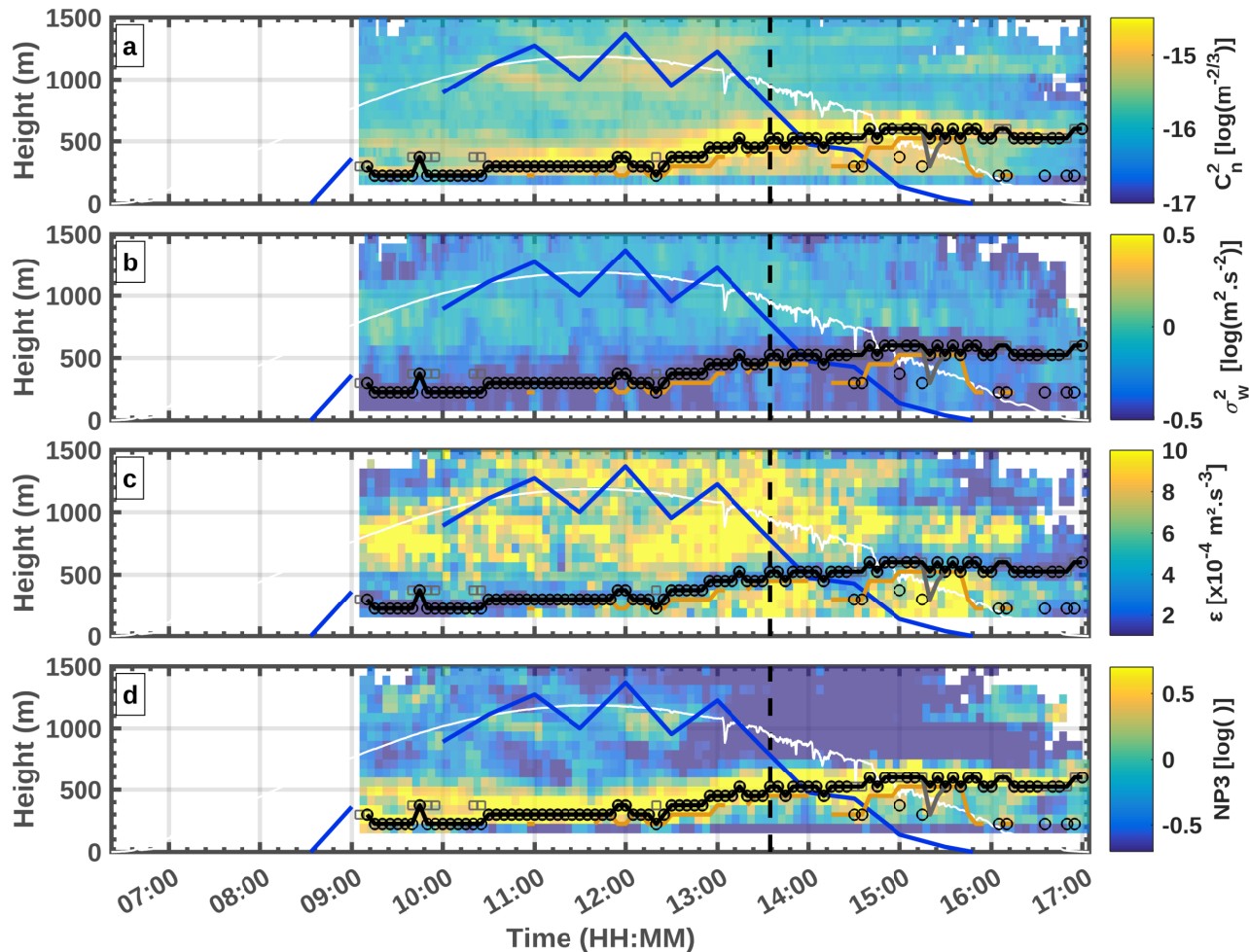

**Figure 4.** UHF RWP observations for 27 October 2021 at P2OA-CRA during clear sky: (a) filtered $C_n^2$ in log scale, (b) filtered $\sigma_w^2$ in log scale, (c) filtered and integrated $\varepsilon$ in log scale, (d) integrated $NP3$ in log scale. For all panels, $Zi$ estimates as described in Sect. 3.2.2 and 3.3.3: $Zi_\varepsilon$ (orange line), $Zi_{NP0_{std}}$ (gray line), $Zi_{NP0_{sup}}$ (gray squares), $Zi_{NP3_{std}}$ (black line), $Zi_{NP3_{sub}}$ (black circles); and based on the same ordinate axis (but with different units): short wave down (W m$^{-2}$) (white line), sensible heat flux (deciW m$^{-2}$) (thick blue line). The vertical dashed line correspond to the time of the discussed radiosounding.

to the CBL top, characterized by a strong gradient of potential temperature and mixing ratio (Fig. 5a and 5b). It also shows that $\sigma_w^2$ (Fig. 5e) and $\varepsilon$ (Fig. 5f) are small at this height, leading to a local minimum. In "ideal" clear days, without external

forcing, we would typically not observe significant turbulence above $Zi$ (Fig. 1e). In this case, forcing is small, with weak wind but the wind shears still generates significant turbulence (Fig. 4c). In a subjective way, we estimate $Zi$ at about 550 m from this radiosonde, where a strong potential temperature gradient is observed, associated with a strong humidity gradient (mixing ratio and relative humidity). This height is in good agreement with all the estimates made by CALOTRITON at that time, and with the simplest standard estimate of $Z_i$ from RWP. This case is a typical clear sky case, with $QF = 1$ for most of the day

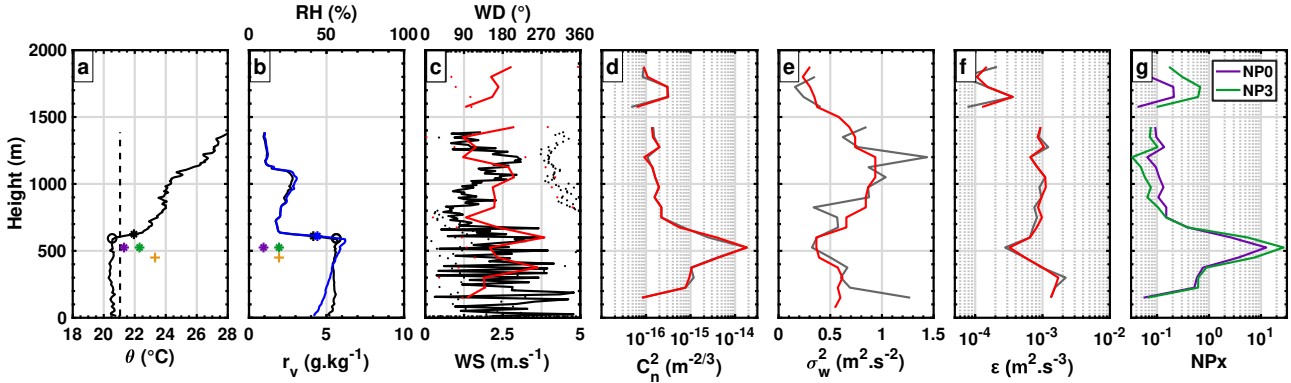

**Figure 5.** Profiles measured by radiosondes and UHF RWP at P2OA-CRA, on 27 October 2021, at 13:35 UTC: (a) potential temperature (black solid line), surface potential temperature + 0.25°C (black dashed line), $Zi$ from in-situ subjective method (black circle), $Zi$ from in-situ potential temperature gradient method (black asterisk), $Zi_{NP0_{std}}$ (purple '×'), $Zi_{NP0_{sup}}$ (purple '+'), $Zi_{NP3_{std}}$ (green '×'), $Zi_{NP3_{sub}}$ (green '+'), $Zi_{\varepsilon}$ (orange '+'); (b) mixing ratio (black line) and relative humidity (blue line), $Zi$ from in-situ mixing ratio gradient method (black asterisk), $Zi$ from in-situ relative humidity gradient method (blue asterisk), purple, green and orange crosses same as described in (a); (c) wind speed (solid line) and wind direction (dotted line) from radiosonde (black) and UHF RWP (red); (d) air refractive index structure coefficient from UHF RWP with raw data (grey line) and filtered data as described in Sect. 3.2.2 (red line); (e) vertical velocity variance from UHF RWP with same colour code as (d); (f) TKE dissipation rate from UHF RWP with same colour code as (d); (g) NP0 (purple line) and NP3 (green line).

(Fig. 4). $Zi_{NP3_{std}}$ has consequently a good confidence index, except around 15:30 UTC, where $Zi_{NP0_{std}}$ is slightly lower than $Zi_{NP3_{std}}$. Note on Fig. 4 that $Zi_{\varepsilon}$ remains equal or below those estimates, and especially decreases in late afternoon, with a strong decay of the surface flux. This is one typical late afternoon transition scenario, as described in Grimsdell et al. (2002) and Lothon et al. (2014). $Zi_{NP3_{sub}}$ also interestingly decays during the same phase, thus defining a potential pre-residual layer, situated between $Zi_{NP3_{sub}}$ (or $Zi_{\varepsilon}$) and $Zi_{NP3_{sup}}$ (or $Zi_{NP3_{std}}$). The pre-residual layer is defined when the

surface heat flux is not strong enough anymore to keep the mixing up to the midday summital inversion, and falls between the thinning turbulence layer and the residual inversion (Nilsson et al., 2016b; Lothon et al., 2023). The different estimates made in CALOTRITON thus can help identify interfaces and layers, in the complex afternoon transition phase. Standard and simple methods do not enable to describe this subtle and still poorly understood complexity.

### 4.2 Cloudy complex case at P2OA

Figure 6 gives another example of UHF RWP measurements on 15 March 2018, this time with a marked external forcing, identified by a cloudy sky and by a high wind speed in the upper layer. For this figure, the cloud base height measured with the ceilometer is added, also revealed by the downward short-wave radiation. In this complex case, the maximum of $C_n^2$ remains most of the day between 2000 m and 3000 m, related with the clouds and associated hydrometeors, rather than to the top of the CBL. This makes $Zi_{NP0_{sup}}$ high in this nearby cloud layer. Between 10:00 UTC and 11:30 UTC, this maximum of $C_n^2$

is competitive with the local maximum below, which is what we can interpret as the top of the growing CBL, and which is better detected with $NP3$. Between 16:00 UTC and 17:20 UTC, the reflectivity field shows the presence of virga (verified by

observations of the weather radars of Meteo-France). Where droplet size is close to the RWP wavelength, this induces a strong reflectivity (and $C_n^2$) on the entire profiles. For this more complex case, $Zi_{NP0_{std}}$, $Zi_{NP3_{std}}$ and $Zi_{NP3_{sub}}$ are consistent only until 11:00 UTC. After this time, $Zi_{NP0_{std}}$ and $Zi_{NP0_{sup}}$ are higher than $Zi_{NP3_{std}}$ and $Zi_{NP3_{sub}}$, suggesting that the latter may be assigned on the top of a TIBL. After 11:30 UTC, the assignments based on $NP3$ become more discontinuous due to the limit of $NPx$ values ($NPx$ profile mean). This discontinuity indicates an increased uncertainty in the attributions. $Zi_{NP0_{sup}}$ is then systematically located above the others, suggesting that $Zi_{NP3_{std}}$ may potentially identify the top of a TIBL. However, we believe that these attributions are correct, as they are located at the height where the strongest wind shear is observed. After 15:00 UTC, Fig. 6 shows more discontinuity on $Zi_{NP3_{std}}$ attributions, demonstrating a CBL complexity with small incoming shortwave radiation, no positive sensible heat flux and the occurrence of precipitation mentioned above.

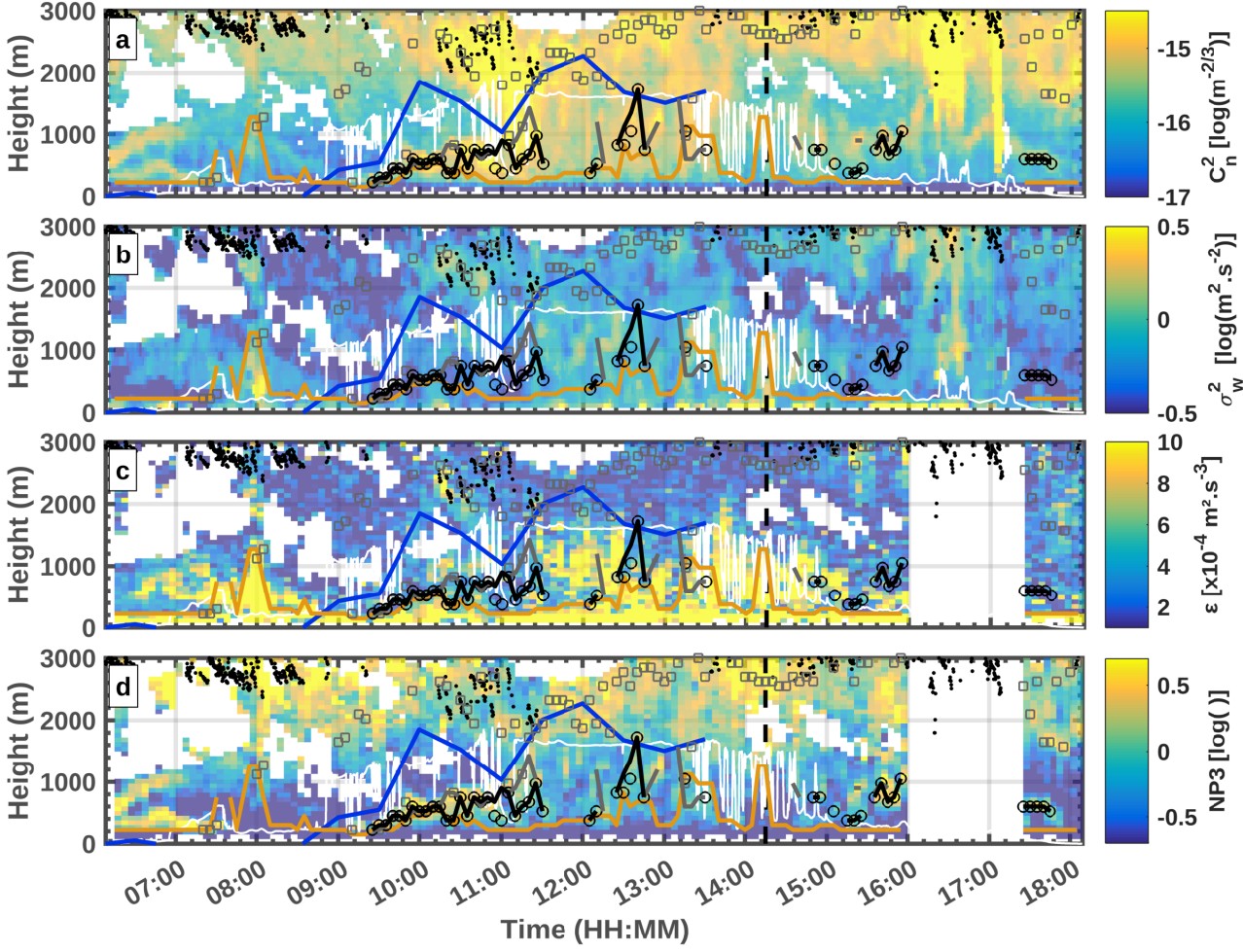

**Figure 6.** LAERO UHF RWP observations for 15 March 2018 at P2OA-CRA with the same description as Fig. 4 and cloud base height measured by CT25k ceilometer (black dots).

In order to better interpret this complex day, Figure 7 compares in-situ measurements of thermodynamical variables with the UHF RWP variables at 14:15 UTC that same day. In a subjective way, $Zi$ can be estimated at 1500 m from this radiosounding

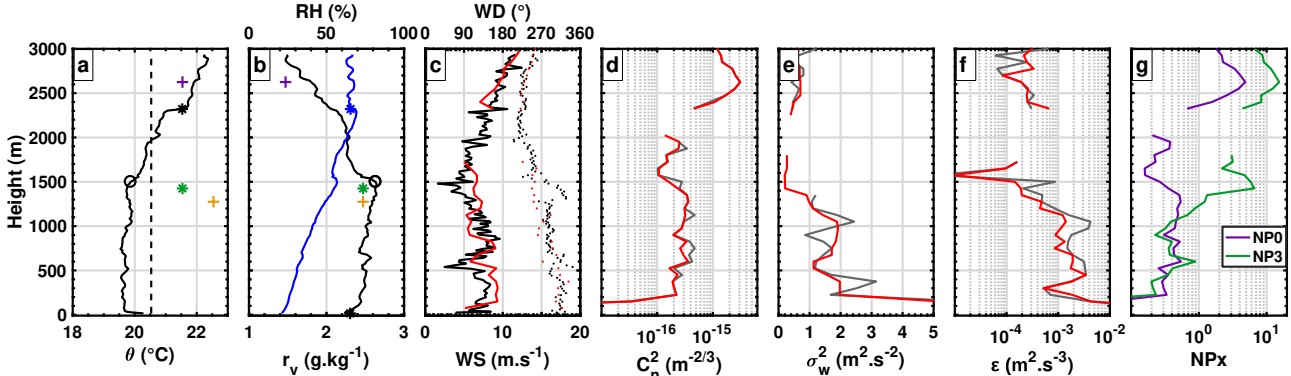

**Figure 7.** Same as Fig. 5 for 15 March 2018, at 14:15 UTC.

(Fig. 7a), the height where the atmosphere starts to be stable (positive $\theta$ gradient), also associated with a strong discontinuity in the mixing ratio profile. This height corresponds well to $Zi_{NP3_{std}}$. Though the absolute maximum of $C_n^2$ (Fig. 7d) and also $Zi_{NP0_{sup}}$ (indicated in Fig. 7a) correspond to an inversion around 2500 m, identified by a strong potential temperature gradient. This actually corresponds to a cloud base (see the black dots in Fig. 6) which is decoupled from the CBL. $Zi_{NP0_{std}}$ is thus unsuccessful here. There is no marked local maximum of $C_n^2$ at the height of $Zi$ estimated from the in-situ radiosonde, but $\sigma_w$ (Fig. 7e) and $\varepsilon$ (Fig. 7f) profiles have a well marked local minimum, forming a marked local maximum on NP3.

This example illustrates the benefit of taking $\sigma_w$ into account via $NPx$ with $x > 0$ in the attribution of $Zi$. It also shows the advantage of the various $Zi$ estimates to identify different interfaces in the case of complex vertical structure. Of course, the large complexity of this case and the weak CBL encountered in some phases of the day due to clouds and precipitation, makes is still difficult to deal with.

## 4.3 Clear sky with multiple layering during LIAISE

The use of the LIAISE dataset (Boone et al., 2021) allows us to test the CALOTRITON algorithm with the same UHF RWP at a different location and under different meteorological conditions. During the LIAISE campaign, the LAERO UHF RWP was deployed from June 2021 to October 2021 in the semi arid region of Lleida, Spain, at a distance of about 15 km from large areas of irrigated crops. Figure 8 illustrates the complexity that can be observed in clear sky conditions in this region, and tests the capability of CALOTRITON for CBL with multilayer conditions. The analyses of this rich dataset have only recently started, but the study by Jimenez et al. (2021) already testimonies to this complexity.

Early in the morning, an elevated local (actually absolute) maximum of $C_n^2$ is present between 2 and 3 km. This corresponds to a high inversion, potentially coming from a residual transported layer (shown later). An algorithm purely based on maximum $C_n^2$ would start the day with this erroneous $Z_i$ estimate. In CALOTRITON, the process of finding the first estimate of the day

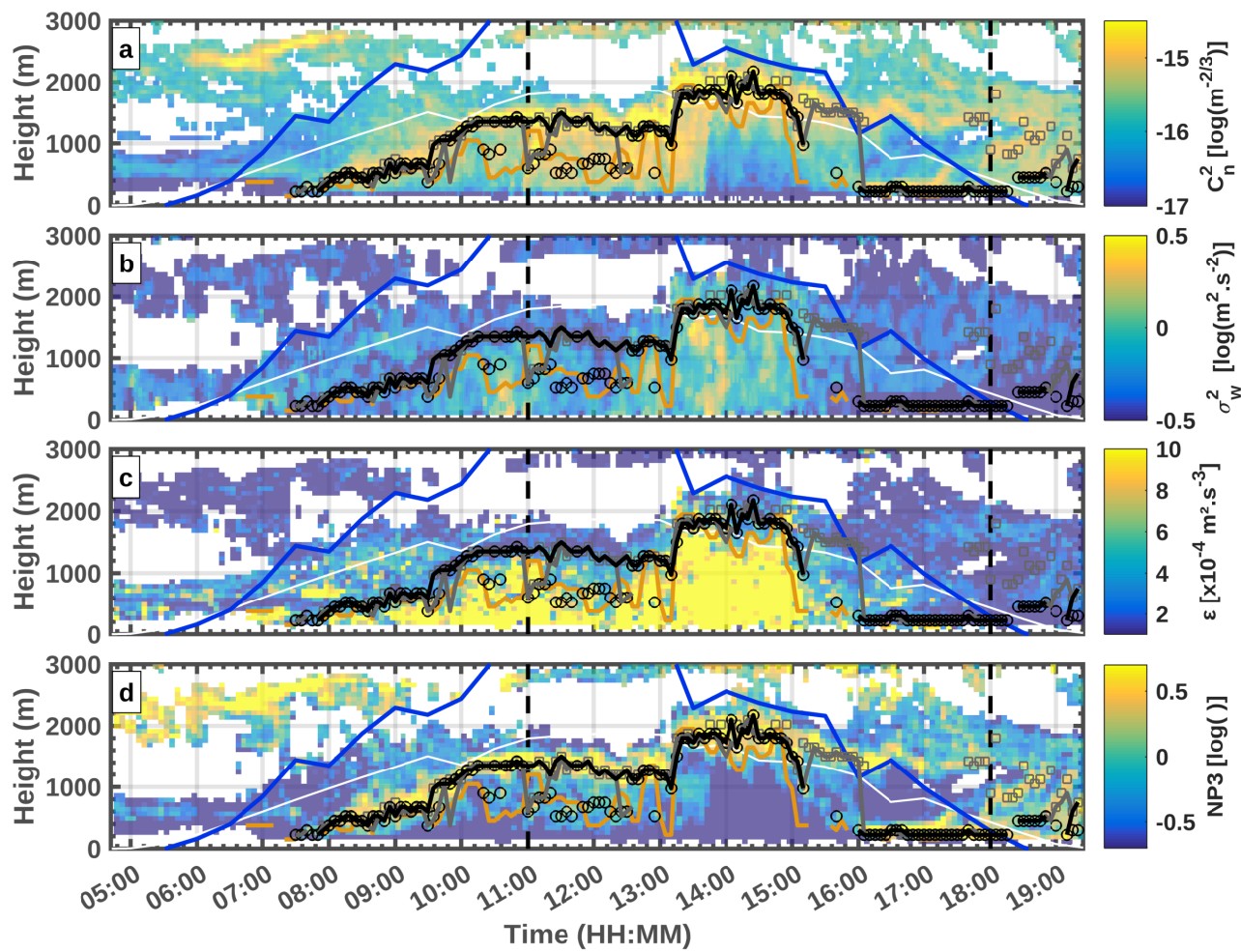

**Figure 8.** LAERO UHF RWP observations for 27 July 2021 at Els Plans (Spain) during the LIAISE campaign with the same description as Fig. 4.

at the first possible gate enables to avoid this situation. Most of the various $Z_i$ estimates agree until 09:30 UTC. Between 10:00 UTC and 13:00 UTC, $Zi_{NP3_{sub}}$ indicates the potential presence of a TIBL located below 1000 m, whilst at 11:00 UTC, $Zi_{NP0_{std}}$ is at the level of $Zi_{NP3_{sub}}$ at about 600 m. Firstly, note the maximum $C_n^2$ discussed previously is still present at 11:00 UTC on Fig. 8a, and corresponds to a large moisture and temperature inversion. It is not thin, but associated with a large change in the water vapour mixing ratio.

Figure 9 shows measurements from a radiosonde taken at this time. In the first 1500 m, we notice the presence of two superimposed layers with constant potential temperatures and mixing ratio (Fig. 9a and 9b), separated by a thermal inversion at 600 m. Strictly speaking, according to the definition of the thermodynamic approach, $Zi$ should be located at the top of the first layer, since the surface over-adiabaticity (28°C) theoretically does not allow a parcel of air to cross the inversion at 600 m

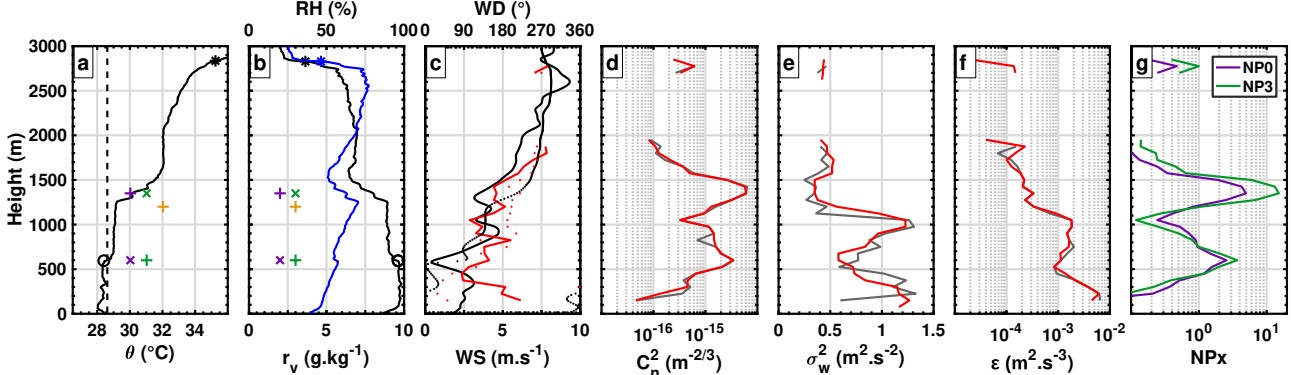

**Figure 9.** Same as Fig. 5, but with profiles measured by radiosounding and LAERO UHF RWP at Els Plans (Spain) during the LIAISE campaign on 27 July 2021 at 11:00 UTC

(29°C above). By a scalar concentration approach, $Zi$ could also be attributed to 600 m where a discontinuity in the mixing ratio is indeed observed. The latter is, however, not considered very strong and the fact that a constant (but slightly different) mixing ratio is observed above and up to 1300 m, indicates mixing within this upper layer. An earlier sounding, at 10:00 UTC

(not shown here), reveals that the CBL was well mixed up to 1200 m a. g. l. this day over this dry site. What is seen at 11:00 UTC on Fig. 9a and 9b is an intrusion of a nearby boundary layer likely advected into the region from the north-east, that is from the irrigated site, which has much thinner CBL. The cooler and moister air observed over the dry site in Figure 8b up to 600 m is consistent with air coming from the irrigated area. In this case, over Els Plans, some turbulence structures may be able to overcome the 600 m high inversion, and some others not. We indeed find high turbulence values ($\varepsilon > 5 \times 10^{-4}$ m$^2$ s$^{-3}$)

up to 600 m. This turbulence contributes to mix both layers and erode the inversion. This is observed later in the soundings (not shown). Comparing the 11:00 UTC radiosonde profile the UHF RWP estimates, $Zi_{NP3_{std}}$ defines $Zi$ at 1300 m, with the presence of a TIBL inside, whose top would be located at 600 m and detected by $Zi_{NP3_{sub}}$ and $Zi_{NP0_{std}}$.

In Figure 8b-c, shortly after 13:00 UTC we notice a sudden increase in turbulence up to about 2000 m a. g. l.. This may be due to another boundary layer advection as the wind direction (not shown) suddenly changes from $\sim 200°$ to $\sim 90°$ between

$\sim 1000$ m and $\sim 2000$ m. A break in the temporal continuity of $NP3$ local maxima is then observed and the imposed growth limit does not allow to follow this sudden evolution. The use of $Zi_\varepsilon$ (1875 m at 13:15 UTC) allows attributions of $Zi_{NP3_{std}}$ and $Zi_{NP0_{std}}$ to follow this rapid change from 975 m at 13:10 UTC to 1800 m at 13:20 UTC. From 14:00 UTC onwards, a low-level marine breeze (< 500 m) can be seen on the Fig. 8a and 8b. This marine air is called "La Marinada" in this region (Jimenez et al., 2021), and is typical of the area. It is an entrance of marine air coming from the Mediterranean Sea, which

is usually favoured by a continental heat low over northern Spain. Between 15:00 UTC and 16:00 UTC, differences between $Zi_{NP3_{std}}$ and $Zi_{NP0_{std}}$ are observed, showing the high complexity of the atmosphere. After 16:00 UTC, all the attributions are made at 225 m on the first UHF RWP gate.

Figure 10 shows the data from a radiosonde launched at 18:00 UTC on the same day, where it can be seen that $Zi_{NP3_{std}}$ and $Zi_{NP0_{std}}$ are well established at the height of the maximum potential temperature and mixing ratio gradient. The observed

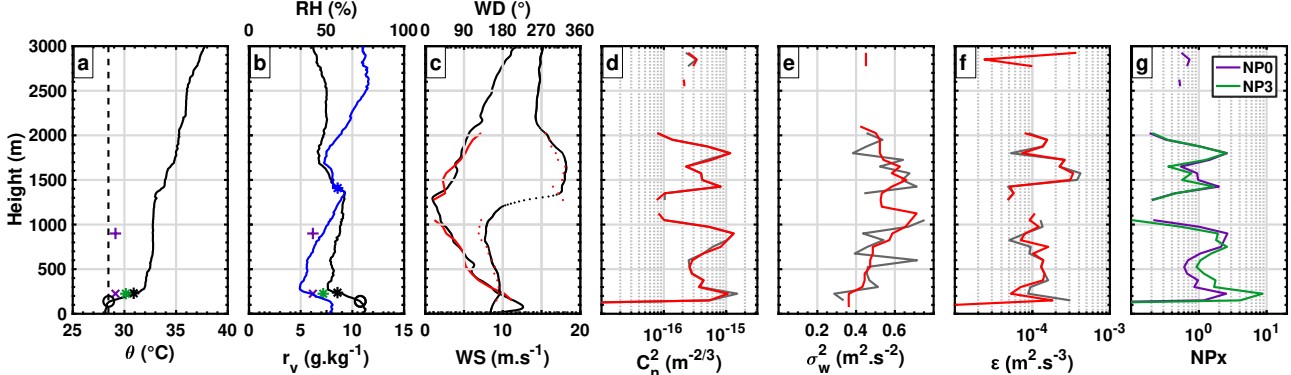

**Figure 10.** Same as Fig. 9 at 18:00 UTC.

breeze has therefore set up a new convective boundary layer. At 19:00 UTC, the radiosonde (not shown) data indicate that $Zi$ decreases below the first reliable RWP gate, CALOTRITON attributions are then erroneously overestimated by about 500 m a. g. l..

This example has shown a highly complex situation, which can occur even in clear sky conditions. It exemplifies the complexity of automatically assigning $Zi$ with radiosonde data or remote sensing, when several boundary layers interact and lead to multilayering of the lower troposphere. It also illustrates how the different CALOTRITON attributions can help identifying CBL top, TIBL top, and the advection of internal boundary layers. The flag defined in Sect. 3.3.3 helps to identify the days when this kind of complex layering of the low troposphere may occur.

## 5   Validation of CALOTRITON with in-situ measurements

The previous sections have shown that $Zi_{NP3_{std}}$ gives the best estimates of $Zi$. To validate this estimate, all CALOTRITON attributions were compared to the numerous radiosonde data made during the LIAISE and BLLAST field experiments, nearby two UHF RWPs (Table 1). During BLLAST, the LAERO UHF RWP was at P2OA-CRA, and the CNRM UHF RWP about 5 km to the South. RPAS (Reuder et al., 2016) profiles were made nearby the two sites, and radiosounding balloons were launched from both sites (Lothon et al., 2014; Legain et al., 2013), a few tens of meters from the RWPs. During LIAISE, the LAERO UHF RWP was installed on a dry area (Els Plans), and the CNRM UHF RWP over an irrigated area (La Cendrosa) (see Sect. 4.3), about 15 km away (Boone et al., 2021). Radiosoundings were launched from the two sites as well, nearby the RWPs (also a few tens of meters). A total of about 500 profiles are available for the evaluation of the CALOTRITON estimates. Median filters are applied over the vertical, to the in-situ data to match a vertical resolution of 10 m. Those numerous in-situ profiles give the opportunity to evaluate and validate CALOTRITON, but also give some insight on the results from automatic estimates from thermodynamic profiles.

In Fig. 11a and Fig. 11b, we compare $Zi_{NP3_{std}}$ with automatic in-situ estimates based respectively on the parcel method (one of the most frequently used), and on the water vapour mixing ratio gradient method (as an example of the gradient methods).

In the parcel method, a small amount $\delta\theta$ is added to the surface potential temperature ($\theta_s$), and $Zi_{parcel}$ is defined as the height where $\theta = \theta_s + \delta\theta$ above surface (Seibert et al., 2000). Here we set $\delta\theta$ as 0.25°C (Fig. 5a). A great disparity of
points is observed, which is mainly explained by a poor estimation of $Zi_{parcel}$ in non-textbook cases. They are indeed either overestimated (example in Fig. 7a), or underestimated by the potential presence of TIBL (example in Fig. 9a). $\delta\theta$ may not be always appropriate, according to the actual super-adiabatism close to surface. In addition, a large number of small $Zi$ estimates by the parcel method ($< 200$ m) can be observed due to the observation of a positive potential temperature gradient in the very first meters of the profiles. Hennemuth and Lammert (2006) attribute this to evening transitions, but it may actually happen at
any time (see Fig. 9a), for example by the establishment of local breezes or other type of advection. It can also occur when the surface layer is not clear (showing fluctuations over the vertical) during the start and at the spot of the sounding. The parcel method may or may not be fair in those cases. The in-situ radiosounding or RPAS profile is very local and instantaneous. Note that using the bulk Richardson method rather than the parcel method did not significantly change the result of this comparison (not shown). The bulk Richardson $Zi$ estimates were actually slightly less relevant than the parcel method estimates, with more
frequent overestimation of $Zi$ due to the attribution of $Zi$ on upper inversions.

The in-situ based gradient methods assign $Zi$ at the height of the strongest gradient of potential temperature, water vapor mixing ratio or relative humidity, below 3000 m. Figure 11b shows the comparison between $Zi_{NP3_{std}}$ and the water vapour mixing ratio gradient estimates. There is a large majority of cases where attributions based on water vapour mixing ratio gradient method ($Zi_{r_v gradient}$) are largely above $Zi_{NP3_{std}}$. They mostly correspond to attributions to residual layers or upper
inversion, as described by Hennemuth and Lammert (2006), and as seen in the previous examples (Fig. 7, Fig. 9). Also a

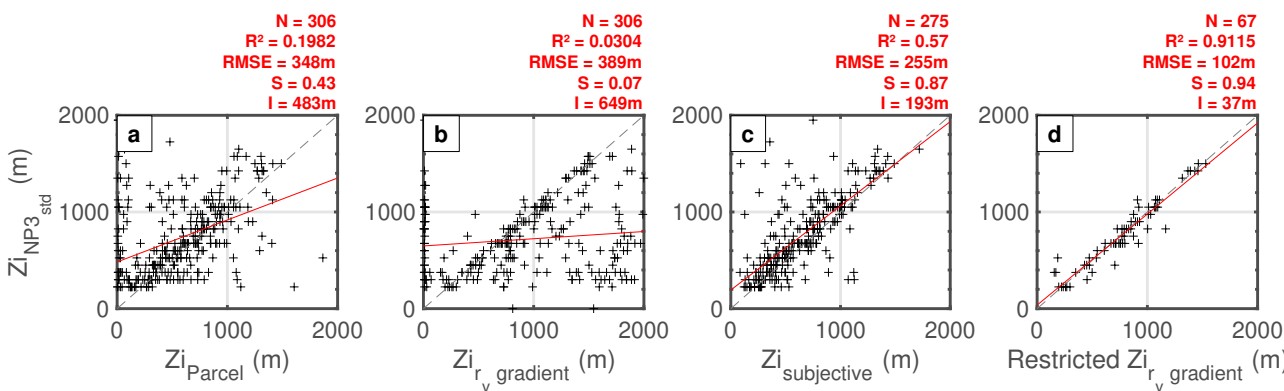

**Figure 11.** Comparison between $Zi_{NP3_{std}}$ and (a) $Zi$ from in-situ parcel method, (b) $Zi$ from in-situ water vapour mixing ratio gradient method, (c) subjective $Z_i$ estimates and (d) restricted $Zi$ from the convergence of all the in-situ-based estimates. In all panels, the grey dashed line represents the 1/1 slope and the red line is the linear regression. The characteristics of the regression are indicated with the red font text: the number of data points (N), the regression coefficient ($R^2$), the root mean squared error ($RMSE$), the regression slope ($S$) and the intercept ($I$).

significant number of attributions by gradient methods are very low and correspond to stable surface layer (around morning or evening transitions), but also to the fact that one can observe large fluctuations in the surface layer, as seen in Fig. 7b. Similar results are found when considering the potential temperature or the relative humidity for the gradient method (not shown).

Figures 11a and b show that it remains difficult to qualify CALOTRITON estimates with the automatically determined estimates from in-situ parcel or gradient methods. For this reason, a subjective method of assigning $Zi$ from in-situ thermo-dynamical profiles is helpful. We attempt to keep this method as objective as possible, by assigning $Zi$ at the height where we observe a first notable discontinuity in the mixing ratio profile associated with discontinuity in the potential temperature profile. The approach is similar to searching for the top of a conserved scalar tracer, and it should also correspond to the height where the entrainment zone starts (see Fig. 1d). Figure 11c shows the comparison of this subjective $Zi$ with CALOTRITON estimates based on $NP3_{std}$. We obtain a much better agreement between the attributions with a higher regression coefficient ($R^2 = 0.57$), but some points still deviate from the trend and may be due to subjective misinterpretation as we have seen in the presence of TIBL for example, or to failure of CALOTRITON estimates.

In order to disregard errors in the in-situ estimates, we finally restrict the $Zi_{NP3_{std}}$ / in-situ comparison to the cases where the standard deviation within the estimates from the various in-situ methods is smaller than 100 m. This way, we ensure consistency between those methods, that is, we keep more "simple" or "textbook" situations. We also ensure objectivity. Figure 11d shows an excellent comparison between $Zi_{NP3_{std}}$ and $Zi$ from the in-situ mixing ratio gradient method in those conditions, with $R^2$ = 0.91 and a root mean squared error ($RMSE$) of 102 m. However, there are still a few points that depart, which are mainly due to:

- late afternoon conditions, when the atmosphere starts to stabilize in the surface layer. In these cases, we are actually at the limit of the CBL definition;

- attributions below the UHF RWP vertical detection limitation.

If we ignore in-situ attributions below 225 m and times later than 16:00 UTC, we obtain $R^2$ = 0.93 and $RMSE$ = 84 m (that is close to the 75 m UHF RWP vertical resolution), which confirms the consistency of CALOTRITON estimates in those conditions.

Table 5 summarises all the comparisons made between the UHF RWP CALOTRITON estimates (based on various orders of $NPx$ in standard configuration as described in Table 4) and in-situ estimates (based on the different methods). $Zi_{NP4_{std}}$ has slightly larger $R^2$ and lower $RMSE$ when comparing with the subjective in-situ $Zi$ estimates. But generally, $NP3_{std}$-based attributions are very similar to $NP4_{std}$-based attributions, and moreover lead to 4% additional attributions when compared to the subjective method in-situ estimates. This further supports the optimum choice of using $Zi_{NP3_{std}}$ to estimate $Zi$ with CALOTRITON, and the validity of those estimates. Finally, when we compare the attributions of $NP3_{std}$ with $QF = 1$ with those of restricted $Zi_{r_v gradient}$ which both reflect simple textbook case, the results are excellent with a $R^2$ of more than 0.96 and an $RMSE$ =71 m, that is lower than the RWP vertical resolution (75 m).

In conclusion, we have also shown that CALOTRITON is not specific to one UHF RWP and one observational site.

**Table 5.** Summary of linear regression characteristics between $Zi$ from CALOTRITON with $NPx$ ($x = 0$ to $5$) in standard configuration as described in Table 4 and $Zi$ from in-situ subjective method, restricted $Zi$ estimates based on in-situ mixing ratio gradient method, agreeing with other in-situ-based estimates as described in the text, and same $Zi$ with further restrictions (no attributions below 225 m or after 16:00 UTC

| Compared $Zi_{NPx_{config}}$ | $Zi_{NP0_{std}}$ | $Zi_{NP1_{std}}$ | $Zi_{NP2_{std}}$ | $Zi_{NP3_{std}}$ | $Zi_{NP4_{std}}$ | $Zi_{NP5_{std}}$ | $Zi_{NP3_{std}}(QF=1)$ |
|---|---|---|---|---|---|---|---|
| | | | | with $Zi_{subjective}$ | | | |
| Number of data points | 288 | 284 | 286 | 275 | 264 | 254 | 142 |
| $R^2$ | 0.43 | 0.56 | 0.47 | 0.57 | 0.59 | 0.56 | 0.62 |
| $RMSE$ | 285 m | 246 m | 309 m | 255 m | 253 m | 270 m | 255 m |
| Slope | 0.74 | 0.85 | 0.86 | 0.87 | 0.89 | 0.89 | 0.9 |
| Intercept | 219 m | 162 m | 204 m | 193 m | 193 m | 200 m | 150 m |
| | | | | with restricted $Zi_{r_v gradient}$ | | | |
| Number of data points | 70 | 70 | 69 | 67 | 66 | 62 | 39 |
| $R^2$ | 0.72 | 0.80 | 0.70 | 0.91 | 0.88 | 0.87 | 0.94 |
| $RMSE$ | 181 m | 149 m | 182 m | 102 m | 117 m | 126 m | 90 m |
| Slope | 0.84 | 0.87 | 0.81 | 0.94 | 0.90 | 0.94 | 0.96 |
| Intercept | 65 m | 68 m | 69 m | 37 m | 76 m | 47 m | 5 m |
| | | | with restricted $Zi_{r_v gradient}$ without $Zi < 225$ m and only before 16:00 UTC | | | | |
| Number of data points | 56 | 56 | 55 | 52 | 52 | 49 | 29 |
| $R^2$ | 0.70 | 0.80 | 0.81 | 0.93 | 0.93 | 0.92 | 0.96 |
| $RMSE$ | 179 m | 138 m | 135 m | 84 m | 79 m | 95 m | 71 m |
| Slope | 0.97 | 0.99 | 0.99 | 1.03 | 1.02 | 1.07 | 1.03 |
| Intercept | -55 m | -38 m | -34 m | -42 m | -33 m | -68 m | -43 m |

## 6 Summary and discussion

With this new algorithm, the main objective of obtaining reliable estimates of $Zi$ with a UHF RWP, for the analysis of long term series, is met, except for CBL thinner than 225 m here.

CALOTRITON uses two surface sensors additionally to the RWP : a humidity sensor at 2 m and a sonic anemometer for the evaluation of the sensible heat flux. We have seen that CALOTRITON can give satisfying results without the sensible heat flux input. The use of the humidity sensor allows to strongly restrict the attributions, especially in the presence of low stratus and fog. It thus remains useful, and a low cost and easy to use input. If this sensor is missing, CALOTRITON will likely attribute inaccurate $Zi$ estimates on the top of the fog, when it occurs. Using the sensible heat flux to restrict the estimations to days with significant fluxes (for example larger than 50 Wm$^{-2}$) can avoid the difficult CBL detection in winter (with very shallow or inexistant CBL), or in certain strong foehn or heat wave cases (when the sensible heat fluxes may be small or even negative).

Relatively to the simpler previously used algorithms for this profiler, and to standard methods, CALOTRITON manages to deal with quite complex cases. Those 'standard methods' are mainly based on catching the appropriate local maximum of $C_n^2$, with help of temporal continuity. In CALOTRITON, the search for the first attribution of $Zi$ at the first reliable UHF RWP gate is a significant progress, consistently with Molod et al. (2015) approach. Also taking into account both the higher

reflectivity at inversions and the amount of turbulence within the CBL by use of the new key variable $NPx$ allows to improve the attributions, in particular in the presence of clouds ($x$ equal 3 or 4 seems the most appropriate). This was also found by

Bianco and Wilczak (2002) with a different innovative 'fuzzy logic' approach.

The criterion of temporal continuity, which appears as a real need, sometimes induces errors. Indeed, the associated jump threshold that is tolerated for the CBL growth is somehow arbitrary, and prevents the potential abrupt growth in certain conditions. Using $Zi_\varepsilon$ to allow larger growth limit in those specific conditions helps to better manage complex cases. This is another improvement brought by CALOTRITON. However, this one also can induce errors, in particular in the morning, by attributing

$Zi$ at the height of residual layers. Using an additional median filter on $Zi_\varepsilon$ could allow us to limit these errors by better considering a certain temporal continuity of $Zi_\varepsilon$. The definition of $Zi_\varepsilon$ could itself be improved. It is by itself an interesting useful variable.

The comparison of CALOTRITON $Z_i$ estimates with in-situ thermodynamic profiles has shown that there is no automatic method based on in-situ thermodynamic profiles which can deal with the complexity of the atmospheric structure, and that the

subjective way remains the best. Such a subjective approach was actually also considered as a reference in Bianco et al. (2008), but applied on the RWP variables.

CALOTRITON is definitely not a simple algorithm, but this actually reveals the need to adapt to the high complexity of the lower atmosphere vertical structure. Bianco et al. (2008) proposed an improved algorithm relative to Bianco and Wilczak (2002), with more complexity added, which demonstrates this need of complexity and adjustments, to optimize the understand-

ing and detection of the appropriate interface. The flag system and various types of $Zi$ estimates proposed in CALOTRITON allow us to express and document this vertical structure complexity, and meanwhile give information on the quality and difficulty of the $Z_i$ estimations. In complex cases, characterizing the convective boundary layer by a single height may actually not be appropriate, in particular in the presence of TIBL where it is difficult to determine (and even define) $Zi$, even based on in-situ thermodynmical data. It becomes very difficult to statistically qualify CALOTRITON attributions in such cases. Over

the 8-year time series of the UHF RWP at P2OA, we find that 17% of the days have more than 75% of their $Zi$ estimates with $QF$=1. This means that about 17% of the days are quite close to textbook cases, with large confidence on CALOTRITON $Zi$ estimates. In contrast, at Els Plans during the LIAISE campaign, none of the days presents $QF$=1 for more than 75% of the time of day. That is there is no simple textbook case in this area during the LIAISE campaign summer.

The use of the different $Zi$ estimates by CALOTRITON is also of large interest for documenting the complex structure of

the CBL, like that found in both P2OA (at the foothills of the Pyrénées ridge) and at LIAISE (with the influence of the sea and the mountain at mesoscale). Though a statistical use should be done only with caution. One can for example estimate the occurrence of significant differences between $Zi_{NP3_{std}}$ and $Zi_{NP3_{sub}}$. At P2OA during the 8-year time series, we find that only 3% of the days show a significant difference between both estimates for more than 25% of the time. This would mean that TIBL are not very frequent at P2OA. In contrast, at Els Plan during LIAISE, we find 26% of such days, which likely means

that TIBL occurs very frequently during the LIAISE campaign. One can also estimate the occurrence of differences between $Zi_{NP3_{std}}$ and $Zi_{NP0_{sup}}$: At P2OA, over the 8-year time series, 72% of the days show a significant difference between both for more than 25% of the time of day. Those days can be related to the large occurrence of cloud layers above the CBL top which

generate an inversion. At Els Plans during LIAISE, this number reaches 92%, which likely means that there are established upper inversions in the LIAISE area. Those preliminary statistics reveal the high complexity of the LIAISE study area, and the

potential of the CALOTRITON various estimates and flags. However, case by case studies and further analyses are needed to help us qualifying this potentiality.

*Code availability.*

CALOTRITON code is available from the authors upon request.

*Data availability.*

Table 6 draws the list of available dataset, with DOI and references. The CT25k ceilometer data are available from the authors upon request.

**Table 6.** Summary of instruments used and datasets

| Instrument | Context | Location | Period | DOI Reference |
|---|---|---|---|---|
| LAERO UHF RWP | P2OA | Campistrous, France | 2015-2022 | Lothon (2023a) |
| LAERO UHF RWP | BLLAST | Campistrous, France | June - July 2011 | Saïd (2012) |
| LAERO UHF RWP | LIAISE | Els Plans, Spain | July 2021 | Lothon and Vial (2022) |
| CNRM UHF RWP | BLLAST | Capvern, France | June to July 2011 | Garrouste (2011) |
| CNRM UHF RWP | LIAISE | La Cendrosa, Spain | July 2021 | Lothon (2023b) |
| CT25k Ceilometer | P2OA | Campistrous, France | 2016-2019 | Contact Author |
| Sonic anemometer | P2OA | Campistrous, France | 2015-2022 | Lohou et al. (2023a, b) |
| Sonic anemometer | BLLAST | Campistrous, France | June to July 2011 | Lohou (2017) |
| Sonic anemometer | LIAISE | Els Plans, Spain | July 2021 | Price (2023a) |
| Sonic anemometer | LIAISE | LA Cendrosa, Spain | July 2021 | Canut et al. (2022) |
| Radiosoundings | BLLAST | Campistrous, France | June to July 2011 | Lothon (2018) |
| Radiosoundings | BLLAST | Capvern, France | June to July 2011 | Legain (2011) |
| Radiosoundings | LIAISE | Els Plans, Spain | July 2021 | Price (2023b) |
| Radiosoundings | LIAISE | La Cendrosa, Spain | July 2021 | Garrouste et al. (2022) |
| RPAS | BLLAST | Campistrous, France | June to July 2011 | Reuder and Jonassen (2017) |

*Author contributions.*

AP is the main author of CALOTRITON algorithm: conception, coding, tests, evaluation, data analysis. He is also the main writer of the article. ML supervised the work and analysis, and helped in the writing. She is the principal investigator of the

LAERO UHF RWP. JA and PYM are the coordinators of the funding contract, and collaborated to the work. BC is the author of the initial code for the UHF RWP data process, and of the previous algorithm for Zi estimates. He helped to the algorithm conception. SD is responsible for the P2OA-CRA instrumentation and data. She and AV helped in instrumentation maintenance,

data process, and data availability. YB operates the LAERO UHF RWP at P2OA and during field experiments, and helped in the operation of the CNRM UHF RWP during LIAISE. FL is the principal investigator of the 60 m tower, and contributed to
the writing. GC was the lead of the instrumental deployment during LIAISE, especially of the CNRM instruments installed at La Cendrosa. JB was responsible for the deployment of radiosoundings at Els Plans during LIAISE, and contributed to the writing. JR was the PI of SUMO RPAS during BLLAST, and contributed to the writing.

*Competing interests.*

The contact author has declared that neither of the authors has any competing interests.

*Acknowledgements.* We gratefully acknowledge the French Atomic Energy Commission (CEA) assisted by the Université Paul Sabatier, Toulouse, for funding this study and their support.

P2OA-CRA observation data were collected at the Pyrenean Platform for Observation of the Atmosphere P2OA (http://p2oa.aero.obs-mip.fr). P2OA facilities and staff are funded and supported by the University Paul Sabatier Toulouse 3, France, and CNRS (Centre National de la Recherche Scientifique). P2OA is a component of the ACTRIS-Fr Research Infrastructure and benefits from AERIS data centre
(https://www.aeris-data.fr/) for hosting service data. The 60 m tower is partly supported by the POCTEFA/FLUXPYR European program.

The BLLAST field experiment was made possible thanks to the contribution of several institutions and supports : INSU-CNRS (Institut National des Sciences de l'Univers, Centre national de la Recherche Scientifique, LEFE-IDAO program), Météo-France, Observatoire Midi-Pyrénées (University of Toulouse), EUFAR (EUropean Facility for Airborne Research) and COST ES0802 (European Cooperation in the field of Scientific and Technical). The field experiment would not have occurred without the contribution of all participating European and
American research groups, which all have contributed in a significant amount (see https://bllast.aeris-data.fr/bllast-supports/). The BLLAST field experiment was hosted by the instrumented site of Centre de Recherches Atmosphériques, Campistrous, France (Observatoire Midi-Pyrénées, Laboratoire d'Aérologie). BLLAST data are managed by SEDOO, from Observatoire Midi-Pyrénées. The French ANR (Agence Nationale de la Recherche) supported BLLAST analysis in the 2013-2015 BLLAST-A project. The french contribution to the LIAISE project has been supported by ANR HILIAISE and Meteo-France. We acknowledge Gilles André, Géraldine Pagan, Vinciane Unger, Alain Dabas,
Alexandre Paci, and GMEI/LISA team of CNRM UMR. We also acknowledge Jeremy Price and all the Met Office team involved in LIAISE.

The contribution of Joachim Reuder to this study was partially funded by the project LOWT, funded by the Research Council of Norway (RCN) under project number 325294.

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
