# Peer review of "CALOTRITON: A convective boundary layer height estimation algorithm from UHF wind profiler data"

_Atmospheric Measurement Techniques, 2023_

## Referee Comment (RC1)

**Referee Review**: *CALOTRITON: A convective boundary layer height estimation algorithm from UHF wind profiler data* by Philibert et al., 2023.

**General Comments**

This manuscript reports on an algorithm to estimate the height of the atmospheric convective boundary layer (CBL) using measurements from a radar wind profiler (RWP). Measurements were taken at two locations for a period of 22 years. The algorithm is validated by comparison with radiosonde observations. The topic addressed by this manuscript falls within the scope of Atmospheric Measurement Techniques since it concerns the use of ground-based observations to estimate the height of the lowest layer of the atmosphere. The manuscript in its present form however has several shortcomings, as summarized below.

The methods used in this work are based on valid physical concepts that have been used extensible to estimate planetary boundary layer heights by many researchers since 1994. Limited results are discussed. The authors state that one of the aims of developing their algorithm is that can be used to obtain a long-term series of daytime estimates of CBL height, yet the manuscript only shows a few days' worth of data, which weakens the argument. The algorithm uses information provided by the RWP measurements and meteorological data to handle most (or as many as possible) conditions that can be encountered in the boundary layer (clouds, precipitation, other interference such as birds, etc.). This approach provides restrictions that may make the automatization of the method rather cumbersome. There are already other algorithms (simpler) that estimate CBL heights from RWP measurements, using backscatter or signal to noise ratio (equivalent to using the air refractive index used here) that have proven to be robust and reliable and have been used with long data records and applied over large geographical areas. In addition, consideration to related work (including appropriate references) are not appropriately given.

The title reflects the contents of the paper, and the abstract provides a concise and complete summary.

This manuscript is not a review paper but a report on a new algorithm to estimate CBL heights from a particular type of radar wind profiler (RWP). With that in mind, I would suggest that the Introduction does not need to be so 'sub-sectioned' as presented. Standard definitions of the height of the planetary boundary layer can be found in textbooks and existing methods to estimate planetary boundary layer height from measurements (of which only a few are mentioned!) can be found in many recent publications, particularly in the most recent review article published in AMT by Kotthaus et al. (2023) and references therein. A few paragraphs in the Introduction, specific to the present article would suffice.

In sum, in terms of scientific significance and quality and in terms of presentation quality, I rate this manuscript as 'fair'.

**Specific Comments**

The authors do indicate their own contribution. The authors do not give proper credits to related work; therefore, I suggest that references be reviewed to address this shortcoming. For example, see Section 2.2.3 and parts of Section 3 of Kotthaus et al., 2023. I note that the latter article is cited (page 23, line 425, in passing, related to what appears to be future work? it is not clear) but it is my opinion that the article is more pertinent to the present work than expressed by the authors.

Page 4, line 75, the authors state "However, this technique is not robust enough for statistical studies based on long series." This is an inaccurate assertion.  (1) If by 'this technique' the authors mean the exact methods/calculations performed by Angevine et al. (1994), then I need to state that I am not aware of any long term (or large geographical extent) study of this kind with RWPs data, which it does not demonstrate that the technique is 'not robust enough' but that such a study has not been done. (2) Using RWPs measurements to retrieve CBL (or PBL) heights with approaches/techniques (or algorithms) that basically follow the same method than that of Angevine et al. (1994) do exits and have shown to be robust and reliable even in the presence of clouds (see for example Teixeira, J., and Coauthors, 2021: Toward a Global Planetary Boundary Layer Observing System: The NASA PBL Incubation Study Team Report. National Aeronautics and Space Administration, 134 pp; https://science.nasa.gov/science-red/s3fs-public/atoms/files/NASAPBLIncubationFinalReport.pdf, and references therein).

The overall presentation of this manuscript is NOT well-structured and clear. Some parts of the paper (text, figures, tables) should be clarified, and others reduced, combined, or even eliminated. See comment about Introduction.

Page 3, bottom: " … existing technique based on Angevine et al. (1994) was used so far for the estimate of Zi with this instrument." What 'existing technique' is being used here? By the authors? applied to the data reported later? Is this the technique reported by Angevine et al. (1994)? Please clarify!

Section 2: A map indicating the locations (lat, lon) of the instruments should accompany Table 1.

Figures 3 and 4 (with text in pages 6 & 7, lines 110-115) show variables that are not defined until 11! Yet they are used to make arguments about comparison. The figures themselves are quite hard to 'read' and follow and not having variables defined make the work of the reader (reviewer) extremely hard!

Figures 2, 4 and 7 are very hard to follow! The dark shading obscures the superimposed line plots and it is hard to follow the lengthy caption in these figures, particularly Fig. 4. The vertical dashed line that corresponds to the time of radiosondes measurements needs to be made clearer in Fig. 2.

Figures 3 and 8: too hard to follow, lines need to be thicker perhaps.  In addition, Fig. 3 uses measurements from two different days – one noted as a clear day and another more complex situation with clouds, etc. Do both days need to be in the same figure? May be making larger panels and separating the two days will help.

Then Fig. 8 caption refers to Fig. 3 ('same as Fig.3 …'); Fig. 3 is on page 7 while Fig. 8 is on page 19!!  The reader is expected to scroll nearly 10 pages to understand and follow Fig. 8?!

Line 345: "From 14:00 UTC onwards, a low-level marine breeze (< 500 m) can be seen on the Fig. 7a and 7b." What exactly indicates in this figure that we are observing a 'low-level marine breeze'?

Figure 9: What significance is given to comparisons with the CBL height computed from thermodynamical variables as measured by radiosondes? Why not use the Richardson number method (or bulk Richardson number commonly used), which is more appropriate for CBL conditions? The results shown in panels a through d in Figure 9 are to be expected and add no meaningful information.

Section 4.2 could be shortened and more concise (and clear!)

The manuscript would benefit from a more defined 'summary and conclusions' section, which rather than discuss initial objectives (repetitive to some extent) would summarize the main findings, contributions, and innovative aspects in this work. As written, all of these are hard to determine.

**Technical Corrections – Minor Comments**

Page 2, line 2: "CBL top (Zi) is a key variable in air quality since pollutants, dust, smoke,... emitted" Is these '…' a typo, an error, of it means more elements? May be better to use 'etc.' or to list items specifically. This is seen again on line #45 on page 3.
Some editing, mainly for clarity in English, will help the text. For example, on line 120, page 7 reads "Figure 3 (panels h to n) confronts in situ measurements of thermodynamical …", perhaps the word needed here is 'compares'.

Line 220: Define CBH. I assume it stands for 'cloud base height' but it needs to be stated.

---

## Author Comment (AC1)

**Reply to Referee #2**

We thank Reviewer #2 for his/her useful feedbacks, which made us improve the manuscript.
We have significantly revised the manuscript.

Below is an answer point by point to the review.

**General Comments**

The methods used in this work are based on valid physical concepts that have been used extensible to estimate planetary boundary layer heights by many researchers since 1994. Limited results are discussed.
It is true the manuscript was lacking of previous works on those techniques. We sigificantly enriched the introduction.

The authors state that one of the aims of developing their algorithm is that can be used to obtain a long-term series of daytime estimates of CBL height, yet the manuscript only shows a few days' worth of data, which weakens the argument.
The few cases are used as illustrative examples. Those are very helpful to highlight the capacity of the algorithm, and the challenges of building an automatic algorithm.
An analyis of the resulting results is not in the scope of this paper. A climotology was indeed realized in Lothon et al 2023, for the characterizatin of the P2OA-CRA site. But is is not the purpose of the present article to make those analyses.
However, many days are actually considered when optimizing the algorithm, or comparing it with in situ profiles.
In the revised article, we made clearer the time period of the used dataset, according to its role.

The algorithm uses information provided by the RWP measurements and meteorological data to handle most (or as many as possible) conditions that can be encountered in the boundary layer (clouds, precipitation, other interference such as birds, etc.). This approach provides restrictions that may make the automatization of the method rather cumbersome. There are already other algorithms (simpler) that estimate CBL heights from RWP measurements, using backscatter or signal to noise ratio (equivalent to using the air refractive index used here) that have proven to be robust and reliable and have been used with long data records and applied over large geographical areas.
In addition, consideration to related work (including appropriate references) are not appropriately given.
Those techniques are indeed very interesting for their simplicity (for many of them) and for their relative robustness. But those would not manage in some complex cases that are pointed here.
All algorithms and approaches face the same issue, in link with the atmospheric vertical structure complexity. And the simplest algorithm can definitely not deal with it.
We have more discussed the existing methods published in the literature, which have common issues and common principle with the algorithm presented here.

This manuscript is not a review paper but a report on a new algorithm to estimate CBL heights from a particular type of radar wind profiler (RWP). With that in mind, I would suggest that the Introduction does not need to be so 'sub-sectioned' as presented.

Standard definitions of the height of the planetary boundary layer can be found in textbooks and existing methods to estimate planetary boundary layer height from measurements (of which only a few are mentioned!) can be found in many recent publications, particularly in the most recent review article published in AMT by Kotthaus et al. (2023) and references therein. A few paragraphs in the Introduction, specific to the present article would suffice.

We had already simplified the introduction structure at the submitting pre-review process.

Here we have decided to keep the two sub-sections of the introduction.

However, we did remove the part relative to the definition of the CBL and more rapidly converged toward the CBL depth measurement techniques.

Before this, we stressed on the importance of this variable, following a suggestion of Rev #1.

We also added several references to the RWP-based Zi retrieval techniques, following both reviewers suggestion.

We cited Kottaus 2023, which we had not in mind when writing our manuscript. We thank the reviewer for suggesting this very relevant article, which we missed during the writing process. Our background discussion now starts with this review paper.

***Specific Comments***

The authors do not give proper credits to related work; therefore, I suggest that references be reviewed to address this shortcoming. For example, see Section 2.2.3 and parts of Section 3 of Kotthaus et al., 2023.

We agree that this was a lack of our submitted manuscript. We did enrich the references in the introduction, and also along the manuscript, when discussing results and in the concluding discussion. Kottaus 2023 is part of the references, and used as an exhaustive review. We notably citet and discussed the work of Heo 2003, Compton 2013, Collaud Coen 2014, Molod 2015, Bianco 2002, 2008, Liu 2019.

I note that the latter article is cited (page 23, line 425, in passing, related to what appears to be future work? it is not clear) but it is my opinion that the article is more pertinent to the present work than expressed by the authors.

This citation of Kottaus et al was not the same article as you mention here. The reference of Kottaus et al 2020 (and not 2023) was actually relevantly made here for the discussion about the STRATFINDER algorithm they developped, and the reasons for differences within the various Zi retrieval estimates.

***Page 4, line 75,*** *the authors state "However, this technique is not robust enough for statistical studies based on long series." This is an inaccurate assertion. (1) If by 'this technique' the authors mean the exact methods/calculations performed by Angevine et al. (1994), then I need to state that I am not aware of any long term (or large geographical extent) study of this kind with RWPs data, which it does not demonstrate that the technique is 'not robust enough' but that such a study has not been done.*

We strickly speaking agree with you here. But we also state that this approach could not be used on large statistical dataset without large errors. This is actually shown by Grimsdell and Angevine, 2002. And the illustrative examples that we give also demonstrate this. Defining Zi as the absolute reflectivity maximum will lead to erroneous attributions, in case of residual layers above, with inversion more marked than the CBL top.

We revised the way to discuss this point (see page 4, lines 90 to 107).

*(2) Using RWPs measurements to retrieve CBL (or PBL) heights with approaches/techniques (or algorithms) that basically follow the same method than that of Angevine et al. (1994) do exits and have shown to be robust and reliable even in the presence of clouds (see for example Teixeira, J.,*

*and Coauthors, 2021: Toward a Global Planetary Boundary Layer Observing System: The NASA PBL Incubation Study Team Report. National Aeronautics and Space Administration, 134 pp; https://science.nasa.gov/science-red/s3fs-  public/atoms/files/NASAPBLIncubationFinalReport.pdf, and references therein).*

We thank the reviewer for the reference. Actually, the publication of Teixeira et al does not specifically address this subject. However, it cites Molod et al 2015 who did applied a "robust" simple algorithm for the statistical study of the composite diurnal cycles Zi in a large area from RWPs (about 30 stations in USA), and over 5 years.

Note, though, that Molod et al 2015 restricted their study to the months of June-July, and find large differences between thosethe Zi estimates made from the in situ radiosoundings based on the bulk Richardson method, and the RWP-based estimates, even on avearge iver five years. Indeed, the bias is about 250 m in average, but can reach more than 700 m at some places (still in average over 2 months and 5 years) – See their Fig. 6 and 7. Also two locations, judged as "complex" are ignored. This means that the point of view is very different in Molod 2015: the idea is to have a global view, with a climatology of the diurnal cycle, and finally work with quite approximate estimates. And the evaluation is made at this scale.

But a finer comparison would reveal the exact same issues encountered in complex – but very common – situations, as shown in our study.

Note, moreover, that Molod et al used a similar approach for an improved estimate of Zi, with a correct start of the CBL growth: they search for an "emergence time" at the first available gate. This is very similar to what we have done. It also chases a local maximum with criteria on the temporal continuity. The main difference is on the key considered variable (SNR in their study, and NPx in our study, which is one innovative aspect brought by CALOTRITON).

For all those reasons, our study and that of Molod 2015 are very complementary, and mutually benefit from each other.

We have citet this reference at several places in the revised paper.

*The overall presentation of this manuscript is NOT well-structured and clear. Some parts of the paper (text, figures, tables) should be clarified, and others reduced, combined, or even eliminated. See comment about Introduction.*

It is true that the organization of the manuscript was not optimized. We have profoundly revised this structure, moved figures, made them clearer, added a table. Several of those changes also answered to Rev#1 specific suggestions.

Introduction was revised, as mentioned earlier.

The description of the context and data was made clearer.

The description of the method was also made clearer, and we removed the illustrative examples from this section.

We made a specific section on the 3 illustrative examples.

*Page 3, bottom: " ... existing technique based on Angevine et al. (1994) was used so far for the estimate of Zi with this instrument." What 'existing technique' is being used here? By the authors? applied to the data reported later? Is this the technique reported by Angevine et al. (1994)? Please clarify!*

We made this sentence clearer in the revised version.

*Section 2: A map indicating the locations (lat, lon) of the instruments should accompany Table 1.*

We do not find appropriate to give maps here. But we improved the explanation of the experimental devices and gave references for precise maps.

*Figures 3 and 4 (with text in pages 6 & 7, lines 110-115) show variables that are not defined until 11! Yet they are used to make arguments about comparison. The figures themselves are quite hard to*

*'read' and follow and not having variables defined make the work of the reader (reviewer) extremely hard!*

We have now moved those figures later in the manuscript, in a dedicated section to illustrative examples, and after all variables are defined and the algorithm explained. We also improved the clarity of the figures.

*Figures 2, 4 and 7 are very hard to follow! The dark shading obscures the superimposed line plots and it is hard to follow the lengthy caption in these figures, particularly Fig. 4. The vertical dashed line that corresponds to the time of radiosondes measurements needs to be made clearer in Fig. 2.*

We have improved the clarity of those figues: we kept only 4 panels over 6, softened the background colours, chose more visible symbols, lines and colors on the front.

*Figures 3 and 8: too hard to follow, lines need to be thicker perhaps. In addition, Fig. 3 uses measurements from two different days – one noted as a clear day and another more complex situation with clouds, etc. Do both days need to be in the same figure? May be making larger panels and separating the two days will help.*

In the revised manuscript, we did separate the two days, so that they are displayed at the same location in the manuscript as their comments.

We also improved the clarity of those figures, by thicker lines and symbols, as suggested by REV #2, and also lighter log grids.

*Then Fig. 8 caption refers to Fig. 3 ('same as Fig.3 ...'); Fig. 3 is on page 7 while Fig. 8 is on page 19!! The reader is expected to scroll nearly 10 pages to understand and follow Fig. 8?!*

In the revised manuscript, we moved the figures later in the text, after the methodology has been explained. Initially, we thought that those illustrating figures were necessary for the understanding of the algorithm. Now, following Rev#1 and Rev#2 suggestions, we moved them after the methodology. They are now in a section which illustrates the capabibily and limitations of the algorithm, with the three cases examples (clear at P2OA, cloudy at P2OA, clear but complex in LIAISE). This way, the figures are well associated to the associated discussion.

*Line 345: "From 14:00 UTC onwards, a low-level marine breeze (< 500 m) can be seen on the Fig. 7a and 7b." What exactly indicates in this figure that we are observing a 'low-level marine breeze'?*

We can see this marine air setting at 14:00 UTC (not shown). It is well seen from the RWP data starting 15:00 UTC, and its associated lower temperature and higher moisture are also well seen on the 18:00 UTC sounding.

This feature is very typical of the area in summer, and is actually called "La Marinada" by the forecasters and local people. We have added a recent reference on this typical local feature (Jimenez 2023).

*Figure 9: What significance is given to comparisons with the CBL height computed from thermodynamical variables as measured by radiosondes? Why not use the Richardson number method (or bulk Richardson number commonly used), which is more appropriate for CBL conditions? The results shown in panels a through d in Figure 9 are to be expected and add no meaningful information.*

For the CBL, the parcel method is one of the most common ones, which fits to the approach we have with CALOTRITON. The gradient approaches also are interesting to consider, since the radar echoe will be very sensitive to the inversions. So it remains interesting to check how those are manifested in the radar signal.

The bulk Richardson method is interesting because it combines the wind gradient and the temperature gradient. But we did not initially consider it as a reference, due to previous experiences where it revealed not to be the most appropriate. It is actually usually used when one also wishes to detect the top of the noctural boundary layer, which is not the topic here.

However, we have tested it more closely following your remark.

Below is a comparison between (a) Zi_NP3_std and Zi_parcel, (b) Zi_NP3_std and Zi_Ri, and (c) between Zi_parcel and Zi_Ri.

[Figure]

This figure shows that there is no big difference between the results of (a) and (b). In other words, using the parcel method or the bulk Richardon method as a reference for the validation of Zi_NP3 leads to the same message. Also panel (c) shows that both methods agree a lot, except that Zi_Ri more often over-estimates Zi (catching upper inversions).

Thus, there is no significant change of our results if we consider the bulk Richardson method rather than the parcel method. We prefered to keep the parcel method as a common reference.

We added a comment on this aspect in the revised version (page 23 lines 476 to 479).

Also our true reference here is the subjective way, because, for now, there is no better method. And if we can manage with it (if the set of data is not too large), it is worth using it.

Our work actually demonstrates that there is no perfect objective/automatic technique from the in situ thermodynamic profiles. And the bulk Richardson method will not make this message different. This is also why for example Bianco et a 2008 also used what they called the "expert" estimates, which are down by eye, from the RWP data, to validate their "fuzzy logic" method.

We have made this point clearer in our discussion of the results

*Section 4.2 could be shortened and more concise (and clear!)*
We have revised this section by removing sub-sections, by simplifying the discussed figure (Fig. 11 in the revised manuscript), and by being more concise. This part is now a section by itself, since we moved the illustrative examples in one separated section (previous section 4.1 becomes section 4, and section 4.2 becomes section 5).

*The manuscript would benefit from a more defined 'summary and conclusions' section, which rather than discuss initial objectives (repetitive to some extent) would summarize the main findings, contributions, and innovative aspects in this work. As written, all of these are hard to determine.*
We have significantly revised the conclusion, and called it "Summary and discussion". We followed Rev#2 suggestion to insist more on the main findings and innovative aspects of CALOTRITON.

**Technical Corrections – Minor Comments**

*Page 2, line 2: "CBL top (Zi) is a key variable in air quality since pollutants, dust, smoke,… emitted" Is these '…' a typo, an error, of it means more elements? May be better to use 'etc.' or to list items specifically. This is seen again on line #45 on page 3.*
This sentence has been removed.

*Some editing, mainly for clarity in English, will help the text. For example, on line 120, page 7 reads "Figure 3 (panels h to n) confronts in situ measurements of thermodynamical …", perhaps the word needed here is 'compares'.*
We have corrected those words, and revised the editing with a Native English research scientist.

*Line 220: Define CBH. I assume it stands for 'cloud base height' but it needs to be stated.*
We have defined this acronym earlier in the text (page 12, line 265)

---

## Author Comment (AC2)

**Reply to Referee #1**

We deeply thank Referee #1 for her/his constructive review which has been very fruitful to improve the manuscript.

We have significantly revised the manuscript in response.

Below is an answer point by point to the review.

*Main comments :*

- The introduction should emphasize more clearly why previous methods from the literature are insufficient and how this new method is addressing the shortcomings. This is done for one specific approach Angevine et al. (1994) but is this really the only one available so for application to RWP data?
  Indeed, we realized that the introduction was too poor in documenting previous works made on Zi retrieval based on RWPs. We have largely enriched it.

- In section 2.1 it should be stated more explicitly which auxiliary data are being used and how. Try to make it clear to a reader who may be interested in repeating this work elsewhere, what type of observations are required in addition to the RWP.

  We have revised the initial Table 1 in order to make the use of the dataset clearer.
  In particular, we changed the directing point of view, and first considered the context (P2OA multi-instrumented site, BLLAST and LIAISE), then the RWP profilers. From there, we considered the auxiliary instrumentation, that was used in the different steps of this study. We specified this use in the table (see Table 1 page 6).

  We also have drawn another table later in the manuscript (Table 3 in the revised manuscript, page 8), with the list of variables needed at the different step. Some of them being optional. The text then gives more details and definitions on those, but at least, the table clearly summaries this information.

- In section 2, separate the description of the different steps: start by introducing data acquisition and pre-processing (e.g. filtering, quality control, averaging), then introduce the calculation of new parameters.
  We have made those aspects clearer in the revised version.
  A first part introduces the measured variables, then we address the data process, and later on, we deal with the data averaging and the calculation of new parameters.

- It is rather unusual to introduce result figures in the methods section. Maybe consider referencing figures that are discussed later but focus on the description of the data in section 2

Yes, that is true. We initially thought that it would help understanding the algorithm, but it actually brought some confusion, some complexity in the paper organisation, and also some redundancy in the manuscript.

We therefore moved those figures to later in the manuscript, in a section dealing with the 3 illustrative examples, all gathered together only after the algorithm was described.

We also decided to separate the initial « Results » section in two new sections : one with those three illustrative examples, and the other with the statistic evaluation of the algorithm with the radiosoundings.   (sections 4 and 5)

- Try to organise section 3, maybe using a table (?) that gives an overview on the input variables to the algorithm with a short comment of the ABL feature they respond to (e.g. mixing at CBL top or rather RL height, etc) and then also a table to the parameters and thresholds, including the definition of tinit with auxiliary observations.

We have made this clearer with the new Table 3 mentioned before (page 8).

- It should be clearly visible how other data such as surface humidity or sensible heat flux are being used and if they are not available at a different sites, what would be the implication of working without such information?

- When the 2 m relative humidity measurements are missing : If there is not such measurement, the algorithm does not detect fog, and may gives Zi estimates during those events. In this case, the estimate is likely to be unaccurate, with attribution of Zi at the top of the fog cloud.

- When the surface flux measurements are missing : tinit is determined from only Cn2 at the third gate or from Ziepsi. Considering surface flux measurements slightly increases the number of Zi estimates (3%), with a few earlier estimates which might not be appropriate, but also a gain in the detection of Zi in certain specific days. The change remains very small, as quantified in the text. (page 13, lines 292-293)

We have specified this better in the final discussion of the revised version (page 25, lines 522 to 526)

- Also when writing, try to reference future sections when appropriate, i.e. at times threshold values are introduced and then explained at a later stage but the reader might not know that further information will be provided later.

We have checked this aspect, and the new organisation actually improved a lot this issue.

- Regarding the flag system in section 3.3.3: this is a very promising approach. You discuss its application based on case studies. However, please also comment on the performance of this automatic characterisation based on a diverse and longer dataset. Could this tool be used as a reliable interpretation of the CBLH results without looking at individual days carefully? If so, it would be interesting, how often the different classes are being detected at your site.

We have discussed the use of the flags and Zi various estimates in a specific point further in the present response.

This is indeed an important point about the potential of CALOTRITON in further documenting the complex structure of the low troposphere.

We made some preliminary statistics on the flags, in order to document the occurrence of textbook situations versus more complex cases. And we also give some statistics on the differences between the different types of $Zi$ estimates. The results are indeed interesting, with for example :

(1) Over the 8-year time series at P2OA :

- 17 % of the days with more than 75 % of the data at flag 1. That means 17 % of days quite close to « textbook cases ».
- 3 % of the days show a significant difference between Zi_NP3_std and Zi_NP3_sub for more than 25 % of the data. This would mean that ITBL are not very frequent at P2OA.
- 72 % of the days show a significant difference between Zi_NP3_std and Zi_NP3_sup for more than 25 % of the data. Those are notably related to the large occurrence of cloud layers above the CBL top which generates an inversion.

(2) During the LIAISE campaign, at Els Plan, the flags show the complexity of the low troposphere :

- 0 % of the days with more than 75 % of the data at flag 1. → There is no textbook case during LIAISE.
- 75 % of the days show a significant difference between Zi_NP3_std and Zi_NP3_sub for more than 25 % of the data. This means that ITBL occurs very frequently during the LIAISE campaign at Els Plan.
- 92% of the days show a significant difference between Zi_NP3_std and Zi_NP3_sup for more than 25 % of the data. → There are established upper inversion in the LIAISE area.

Even if those results need to be taken with caution, they interestingly reveal the higher complexity of LIAISE study area, with consistent result. Case studies will help better qualifying the potential of this flag system.

In the concluding discussion of the revised manuscript, we give those intersting examples of statistics to illustrate the potential of using the flags and Zi estimates for the interpretation of the low troposphere vertical structure, and occurrence of elevated inversions and ITBL (page 26-27, lines 552 to 568), but also point on the need to better qualify this potentiality before generalization.

- When discussing the radiosonde comparison in Section 4.2, please reference literature on how you interpret the diversity in results from different radiosonde methods. Maybe it would be more conclusive to work with the subjective method only for the evaluation of the new algorithm and move the other scatterplots to the supplement material? It is not clear what we learn from looking at all the different results.

Since we still find interesting to show the issues raised by the different automatic estimates from in situ thermodynamic profiles, we decided to find a compromise between your suggestion, and the original proposition. Thus, we kept the parcel method and only one of the gradient method, and then showed the subjective method and the selection (« converging ») set (Fig 11 page 23). This enables us to enlarge the figure, and avoid redundancy without loosing the messages. We also made those messages clearer, about what we learn from those comparisons (page 23).

- Figures 2, 4, 7: Please consider a different way to present the data. The figures are very noisy and it is difficult to find the relevant information. Maybe try a different colour scale for the shading in the background? And reduce the amount of layer heights that are being shown? Choose symbols that are always clearly visible, even when several layer heights agree, i.e. they should not overlay each other so that they are not visible. Is it essential to always show all 6 panels? Maybe some could be moved to a figure in the supplement material?

It is true that those figures were complex. We used some different colors in the back for the height-time section, with some transparency. We also changed the line colors and the symbols, to have them all more visible. Finally, we here decided to remove the windspeed and wind direction panels. The wind is useful to understand the structure and estimated Zi (shear for the first example, strong wind for the second, circulations for the third), but it actually is visible on the isolated radiosounding profiles shown in other fugures. We thought those may be sufficient.

Former figures 2, 4, 7 have accordingly been changed  (now Figures 4, 6, 8 in revised manuscript).

- Figure 6: please add mean and median or other statistics in the figure to facilitate comparison of the barcharts.

We have added this information on former Figure 6 (now Fig. 3).

*Minor comments :*

P2, l26: The motivation paragraph of the introduction is very short. Might be useful to add a few more aspects that highlight why the height of the CBL is a variable that should be better characterised based on observations. Maybe move lines P3, L67-71 to beginning of the introduction, i.e. why is CBLH important for studies in complex terrain atmosphere dynamics?

We have re-orientated the start of the introduction toward the importance of estimating the CBL depth, and enriched it (page 2 lines 23 to 29). However, we kept the lines P3, L67-71 of the former manuscript there, because they motivate the monitoring of the CBL depth on this specific P2OA-CRA site, where we're making the long term series.

P2, Introduction: While it is important to highlight the diversity of techniques that are being used to measure the CBL height, it could be useful for this paper to highlight especially the shortcomings of previous methods applied to wind and turbulence measurements. i.e. demonstrate why a new algorithm is needed. And maybe provide some insights on the strengths of UHF input data compared to other measurements.

We have discussed those points, see page 4-5 lines 90 to 127. We notably citet and discussed the work of Heo 2003, Compton 2013, Collaud Coen 2014, Molod 2015, Bianco 2002, 2008, Liu 2019.

P3, Introduction: Please have a look at the recent review on ABLH detection measurements conducted by the PROBE COST action: https://doi.org/10.5194/amt-16-433-2023

We were not aware of this very interesting and well made review when we wrote the article, due to time concomitance. This is a quite timely review. As an exhaustive review, we took it as a starting reference in the revised manuscript.

P4, L85: "CBL height"

Typo corrected.

P4, L85: Is the time series 22 years long? On Page 3, line 67 it is stated that the UHF observations started in 2010. And according to table 1 observations started in 2011. Please give consistent information.

It is true this is not easy to follow. 22 years is the total length of this UHF RWP time series at P2OA-CRA (with some breaks when it is moved to external field campaigns). But only the 2015-2022 time period was used for the algorithm developpement. 2018 is a year taken for the configuration optimization (common, year with the CT25k ceilometer). 2011 is the year of the BLLAST field campaign.

We made all this clearer in the manuscript.

P5, l101: Please comment on the maximum range with a good confidence level.

We added this comment (page 6, lines 156 to 162), and added this specification in table 2.

The maximum height for this mode is usually around 3 km a.~g.~l., but may be only 500 m or 1000 m in winter, when dry anticyclonic conditions occur. It can reach 7 or 9 km within deeper clouds and rain. For the detection of Zi, we limited the search within the first 3000 m, which we made clearer in the revised version (page 8 line 194).

P5, l104: Is there a name for the retrieval of the 3d wind from the radial velocity? Or at least a reference? Is this done with an internal algorithm by the radar or does it requires post-processing? Are the data filtered for noise? Or any other quality control applied?

The 3D wind retrieval is the typical Velocity Volume Processing method. We specified it more precisely. It is used here with an internal algorithm (improved relatively to the manufacturer software), as well as the preceding step of selecting the meteorological spectral peak. Unfortunately, this procedure is not specifically published, but used in many studies and associated publications. We refered to one of them (page 7, lines 164 to 168).

P5, l110: Please provide reference for aperture correction. Even if it is the manufacturer's user manual.

Thank you for pointing this. It is not a manufacturer source. We used the same technique as Jacoby-Koaly 2002, and so refered to them in the revised version (page 7, lines 173-174).

P5, l111: What is the distance between the radar location and the radiosonde launch site?

It is only a few tens of meters each time. We made this clearer in the revised version (page 22, line 457) . We did it by making clearer the position of the RWP and colocated radiosoundings, for each field experiment.

P11, l154: no plural for "fog"

Typo corrected.

P11, L161: averaging across how many gates?

We specified this (page 10, line 215 ). The averaged is made up to 3000 m maximum.

P11, L116-172: Maybe better structure the method description into data preparation (such as averaging and cleaning) and then start to introduce the new dimensionless variable and the detection procedure.

Effectively, that makes things clearer. We have made it clearer the data process upstream of the new parameter calculation. We actually added a sub-section 3.2.2  for this (page 10).

P11, L177: Please reference literature that describes the use of TKE or dissipation rate for the detection of the CBL height. Has your approach been used before? Same threshold? Also using UHF profiler data as input?

There is no dedicated publication for this Zi retrieval technique. But we did use this technique before and described it in the corresponding publications. We commented on this point more precisely and citet those references (page 3, lines 57-58 ).

P13, L236: Why is it important to give more weight to Cn2 rather than sigma_w?

Cn2 remains « the lead » for the Zi retrieval, that is the sommital inversion must have a significant weight.

The subsequent search for the optimum order 'x=3' confirms this.

P17, L323: Any studies that can be referenced regarding the complexity of ABL dynamics in the study area in Spain?

LIAISE is a pretty recent field experiment, and the complexity is still more revealed through the analysis of its dataset, but one recent publication, associated to LIAISE, can already testify to this

complexity (Jimenez 2023, Mangan 2023). We cite it in the introduction of the third illustrative example (page 19, line 408).

P19, L354: Comment on how reliable the flag system is expected to be over a longer time period? Do you consider it useful for interpretation of cases studies or is it in fact a useful tool to automatically characterise the complexity across several years of measurements? How can this be assessed?

We discussed this point in the concluding discussion (pages 26-27, lines 550 to 568).

- We first consider flag as index of complexity, and reliability of the « best » Zi estimate. Making statistics on the flags is possible, to assess the occurrence of textbook cases versus complex cases.

- But we also consider the flags and the various estimates of Zi definitely useful for case studies. We are starting to use them in the context of the LIAISE experiment, where many complementary observations can help us understand the complex 3D and multi-processes situation.

- What is not easy to say and assess, is the possibility of further interpretating the flags and Zi estimates (with definitions of multi-layers like ITBL and RL) from a statistical point of view.

The case by case studies mentioned before is a very useful step to estimate how the interpretation may be generalized, for a statistical use of the flags and « Zi » various estimates.

P20, L364: How did you choose the appropriate method and why? Please reference literature and comment on the uncertainty that this could have on the comparison results.

We did not intend to be exhaustive on those methods, but wished to consider some methods that were relevant enough for the comparison with the RWP measurement point of view.

We actually chose to try several typical methods, knowing that :

- the parcel method had proved satisfactory in previous experiences, and has common aspects with a search of turbulence layer

- the gradient methods may very well correspond to some local maximum of $Cn2$

- the subjective method is always very useful for the precise understanding of what is happening.

We did omit the relative humidity maximum which though proved to be useful in some past experiences, but was not more relevant than the previously mentioned here.

We also omitted the bulk Richardson method, in spite of its large use. The reason is that we had experienced that it was not always appropriate. However, this method could appear relevant, due to its combination of temperature gradient and wind gradient. We did estimate it in response to Referee #2, and verified whether it was more appropriate. But we found that using this technique was nit better than using the parcel method, and did not change our message on the automatic detection methods from in situ profilings, nor on the evaluation of CALOTRITON.

We probably missed several of them, less commonly used.
We added some discussion about this aspect (page 23, lines 474 to 479 and page 24, lines 484 to 487).

P20, L378: What is the physical meaning of your CBLH is a stable layer is advected near ground level below?

It is true that as soon as an internal stable (or even simply different) layer is advected within the current CBL, then this CBLH looses its meaning and « status », turning from CBLH to residual layer top…

We made this point clearer in the revised manuscript (page 23, lines 473-476), by simplifying the formulation.

It is one reason why the use of Zi_NP3_sub is interesting. One can keep monitor the previous CBL, and change to RL, and have an idea of the entrance of an IBL within, and of its depth (whatever its internal stability characteristics).

P24, L465: Yes, you can analyse a long time series but with the current assessment restricted to convective conditions omitting shallow boundary layer development in winter.

Right, we corrected this in the conclusion. The latter has been significantly modified, following Referee #2 comments. But this point is made at the start of the conclusion (page 25, lines 519 to 521..

---

## Author Response (AR2)

**Cover Letter**

Dear Editor,

We thank you and the reviewers for your feedbacks.

We would like to note that we have answered as much as possible to Reviewer#2 concerns. However, although stated as « major », the comments of Rev#2, as they were formulated, did not allow us to precisely understand what should be modified for the article for it to be significantly improved.

Only one comment seemed indeed major, and could have actually been pointed at the first review: the usefulness of the three illustrating examples.  For this point, we are quite convinced that those examples are indispensable to the understanding of how the algorithm works, and make it more accessible and concrete than without. So, we did not wish to remove them. The manuscript would lose a lot of clarity and interest without those examples.

We therefore made a revision that corresponds to every formal point that seemed to need clarification or stress.

To answer to Rev#1 single comment (only addressed to the Editor) about Figure 4 (and 6, 8), we changed the symbols to make them clearer.

We hope those changes will enable this manuscript to be accepted.

Best regards,

Alban Philibert and co-authors.

**Reply to Reviewer #2**

**General Comments**

This manuscript reports on an algorithm to estimate the height of the atmospheric convective boundary layer (CBL) using measurements from a radar wind profiler (RWP). Measurements were taken at two locations for a period of 22 years.

→ Note that, as mentioned in the previous reply on the same statement of Rev#2, the sentence « Measurements were taken at two locations for a period of 22 years » is not correct.

We had already given in the first reply the following statements:
*"22 years is the total length of this UHF RWP time series at P2OA-CRA (with some breaks when it is moved to external field campaigns). But only the 2015-2022 time period was used for the algorithm development. 2018 is a year taken for the configuration optimization (common, year with the CT25k ceilometer). 2011 is the year of the BLLAST field campaign.*

*We made all this clearer in the manuscript. "*

The manuscript indeed clearly specifies what period is used at one or the other location, and for what purpose (see Table 1).

The algorithm is validated by comparison with radiosonde observations. The topic addressed by this manuscript falls within the scope of Atmospheric Measurement Techniques since it concerns the use of ground-based observations to estimate the height of the lowest layer of the atmosphere.

The title reflects the contents of the paper, and the revised (and improved) abstract provides a concise and complete summary.

The methods used in this work are based on valid physical concepts that have been used extensible to estimate planetary boundary layer heights by many researchers since 1994. Illustrative (limited) results are discussed in detail. The algorithm uses information provided by the RWP measurements and meteorological data to handle most (or as many as possible) conditions that can be encountered in the boundary layer (clouds, precipitation, other interference such as birds, etc.). This approach provides restrictions that may make the automatization of the method rather cumbersome.

→ It is true that several criteria are considered in order to optimize the detection of the convective boundary layer, and properly detect the various interfaces of the low atmosphere in case of complex situations. For now, there is no easy and straightforward algorithm that can disentangle this complexity.

There are such a variety of situations, and many of them can be so complex that it is already very interesting (and a clear advancement) that this algorithm helps in detecting such complex cases and in attributing the nature of each detected interface. Finding it « cumbersome » seems to be a judgement that does not lead to a proposition of improvement of the algorithm or manuscript. Contrary to this, one can find the algorithm complex, reflecting the complexity that it attempts to catch.

The authors have responded to reviewers' comments of their initial submission by significantly improving the manuscript with respect to structure and proper attribution to previous (pertinent) works, including more discussion of those works. I note in particular the clarity and conciseness of sections 2.1 and 2.2.

→ We thank the reviewer for acknowledging this revision work.

The authors in the revised version indicate their own contribution more clearly.

→ The revised version is the same as the submitted version for this specific point. Contributions were already clear at the submission stage.

The lengthy discussion section about flags is hard to follow, perhaps it can be made more concise.
→ Flags are defined in section 3.3.3, lines 324 to 337 p.14-15. There are actually only defined here, with the minimum of information, that is necessary to understand them. We do not think this part is too long and hard to follow.
Maybe the « lengthy discussion » that Rev#2 mentions here is the previous paragraph that defines the various estimates made with the algorithm. (Note that it is also a definition, rather than a discussion.) Those different estimates enable us to detect the complexity of the low layer, and are used to define the flags. The simpler the low troposphere (no multiple interfaces detected), the smaller the flag, the higher the confidence in the estimate of the CBL top. **We have revised the wording of this part, and simplified the sentences**. But we kept the level of information, which is the minimum information for a precise definition of all estimates and their interpretation as main or secondary interfaces.

Figures have been improved. Some aspects of the figure are still hard to follow.
Without any precise remark on the difficulty to read the figures, how to still improve them, and which figures are concerned, we did not change them. **However, we took account of Rev# 1 suggestion on making the symbols of Fig. 4 clearer, by changing them from '+' signs and crosses to squares and circles, respectively.**

Figure 4 is surprising and may merit further scrutiny; why the CBL height diurnal cycle does not show a more canonical behaviour given that "The shortwave radiation shows that this day was mainly clear, with only a few thin and occasional cirrus clouds in the afternoon."?
→ This example actually shows that even in clear air, there may be conditions that lead to non-canonical CBL. This is very common though, especially at P2OA near the Pyrénées mountains. For example, heat waves, or foehn situations, the surface sensible heat flux can be very small, or even negative during daytime, leading to very thin CBL. Also in clear sunny days, convection over the all range of the Pyrénées ridge can lead to a significant subsidence in the foothills area. Where P2OA-CRA site is, which prevents the CBL to deeply grow (Pietersen et al 2014). The example shown in Figure 4 is maybe not a « textbook » case, but remains typical of the area. This case also remains a clear sky case, which shows the classical retrieval of Zi top from Cn2 maximum (« typical » and « textbook » was also used for this aspect in the text). **We have modified the text to avoid confusion on « textbook » terminology, and removed this term. We also have discussed in the conclusion about the complexity of the CBL in the Foothills of the Pyrénées.**

The discussion of the three cases mainly based on these figures is cumbersome and to some extent not completely justifiable with these results. Specifically, the existence (or evidence for) of "pre-residual layers" and the possible identification of the top of turbulent internal boundary layers are at best 'indicated' in these data.

→ We do not agree with Rev#2 on this point. The three examples are key illustrations of the way the algorithm works, and of its ability to deal with the complex vertical structure of the low troposphere. Without those examples, the reader would not be able to easily catch the complexity we are talking about, and the various aspects that we are taking into account like clouds, multi-interfaces (or multi-layering), afternoon and morning transitions, ...

About the pre-residual layer, note that this is a result that is coming from the studies associated with the BLLAST field experiment and project (Lothon et al 2014), which took place on the same site in 2011. Cited references are given for more details.
It is the idea of the various estimates given in CALOTRITON to be able to follow this transition phase, with both a residual inversion and a descending top of turbulence layer, before the surface layer is stabilized.

It remains unclear to this reviewer that the algorithm proposed here can be applied easily to other

locations with only a radar wind profiler available and/or that it will yield the seemingly rich results discussed at length in the cases presented.

By showing the results in a totally different area and forcing, we show that the algorithm can be applied at other locations. It is clearly stated in section 4.3, lines 404-405 p.19, that the LIAISE example illustrates the test of CALOTRITON on a different location. Moreover, the chosen area of LIAISE actually reveals a high complex boundary layer structure that CALOTRITON is able to deal with.

This is also stated on lines 448-452 p.26 of this section. Line 518 p.26 also states: « In conclusion, we have also shown that CALOTRITON is not specific to one UHF RWP and one observational site. » In the conclusion, lines 569-571 p.27, are also related to this aspect.
**We have nevertheless further commented in the conclusion about the complexity of the CBL at both P2OA (foothills of the Pyrénées) and in LIAISE area.**

We have also showed in this article that if limitations of the capacity of CALOTRITON arise, there are mainly due to the high complexity of the low troposphere itself, like what can be found in complex terrain. But this would not be intrinsically due to the algorithm. The flag system and various estimates though, enable us to identify the complex situations, and to characterize them.

**Technical Corrections – Minor Comments**

Some editing, mainly for clarity in English, will help the text.
**We have had the manuscript read again carefully, a few spelling mistakes have been identified and corrected. The document showing the differences between the versions helps to identify them clearly.**